# Learning Vision and Language Concepts for Controllable Image Generation

**Shaoan Xie** [* 1]   **Lingjing Kong** [* 1]   **Yujia Zheng** [1]   **Zeyu Tang** [1]   **Eric P. Xing** [1 2]   **Guangyi Chen** [1 2]   **Kun Zhang** [1 2]

## Abstract

Concept learning seeks to extract semantic and interpretable representations of atomic concepts from high-dimensional data such as images and text, which can be instrumental to a variety of downstream tasks (e.g., image generation/editing). Despite its importance, the theoretical foundations for learning atomic concepts and their interactions, especially from multimodal distributions, remain underexplored. In this work, we establish fundamental conditions for learning atomic multimodal concepts and their underlying interactions With identfiability guarantees. We formulate concept learning as a latent variable identification problem, representing atomic concepts in each modality as latent variables, with a graphical model to specify their interactions across modalities. Our theoretical contribution is to provide component-wise identifiability of atomic concepts under flexible, nonparametric conditions that accommodate both continuous and discrete modalities. Building on these theoretical insights, we demonstrate the practical utility of our theory in a downstream task text-to-image (T2I) generation. We develop a principled T2I model that explicitly learns atomic textual and visual concepts with sparse connections between them, allowing us to achieve image generation and editing at the atomic concept level. Empirical evaluations show that our model outperforms existing methods in T2I generation tasks, offering superior controllability and interpretability.

## 1 Introduction

Concept learning seeks to extract semantic and interpretable representations from high-dimensional data, such as images and text. These representations are essential for a wide range of machine learning tasks, including visual understanding and generation, transfer learning, and explainable decision making (Gal et al., 2022a; Jahanian et al., 2019; Härkönen et al., 2020; Shen et al., 2020; Wu et al., 2021; Ruiz et al., 2023; Burgess et al., 2019; Locatello et al., 2020; Du et al., 2022a;b; Liu et al., 2023). For example, learning *atomic* concepts within both visual and textual modalities, along with their patterns of interaction, facilitates seamless alignment between these modalities. This alignment can significantly enhance downstream tasks such as controllable text-to-image (T2I) generation and editing.

However, achieving this alignment without proper guiding principles is highly challenging. In practice, current T2I methods often suffer from controllability issues, where modifications specified in the text input lead to unintended variations in the generated images (Saharia et al., 2022; Ding et al., 2022; Esser et al., 2024; Liu et al., 2022; Lee et al., 2023; Liu et al., 2024; Betker et al., 2023). These issues stem from the entanglement of multimodal concepts and the resulting misalignment. These empirical challenges underscore the need for principled approaches to concept learning. Unfortunately, the theoretical foundations in this area have been lacking. Only recently, Kong et al. (2024) have formalized concept learning within the framework of latent-variable identification, proposing identification conditions for learning discrete hierarchical models. Nevertheless, their theoretical formulation is limited to single-modal distributions, restricting its applicability for aligning concepts across different modalities as required in T2I applications.

In this work, we seek to establish fundamental conditions for *learning atomic concepts and their interactions from multimodal distributions*. We approach this challenge by framing it as a latent variable identification problem, where atomic concepts and their interactions are represented as latent variables and a graphical model among them. Under this framework, our goal can be formalized as:

*Under what conditions can we identify multimodal latent variables and the underlying graph?*

Closely related to our problem is the literature on multimodal causal representation learning (Yao et al., 2024; Morioka & Hyvarinen, 2023; 2024; Daunhawer et al., 2023; Sturma et al., 2023; Gresele et al., 2020; Sun et al., 2024;

*Equal contribution  [1]Carnegie Mellon University  [2]Mohamed bin Zayed University of Artificial Intelligence.

*Proceedings of the 42^{nd} International Conference on Machine Learning*, Vancouver, Canada. PMLR 267, 2025. Copyright 2025 by the author(s).

von Kügelgen et al., 2021). In particular, Daunhawer et al. (2023); von Kügelgen et al. (2021); Yao et al. (2024); Gresele et al. (2020) have demonstrated that a representation block can be identifiable if it is shared across different modalities. However, these block-wise identifiability results are limited to the shared structure blocks, which may still encompass mixtures of multiple atomic concepts. This limitation hinders their applicability for tasks requiring precise control over individual concepts, such as controllable text-to-image (T2I) editing, a challenge our work sets out to address. While Morioka & Hyvarinen (2024) and Morioka & Hyvarinen (2023) have achieved component-wise identifiability, their approaches rely on semi-parametric assumptions about the latent distributions, including the use of the exponential family and assumptions of additive causal influences. These conditions may be too restrictive for modeling the complex, high-dimensional image-text distributions encountered in real-world applications. Furthermore, Sun et al. (2024) offer component-wise identifiability for multimodal representations by leveraging sparse connections between modalities without specific parametric assumptions. However, their method depends on particular sparsity conditions involving Jacobian matrices of functions among continuous variables, making their theoretical framework incompatible with modalities that include discrete variables, such as text.

In this work, we establish component-wise identifiability for atomic concepts under flexible nonparametric conditions that accommodate both continuous and discrete modalities. Table 1 highlights the key distinctions between our approach and existing theoretical frameworks. Specifically, we assume that the paired text $\mathbf{t}$ and image $\mathbf{i}$ are transformations of underlying atomic textual concepts $\mathbf{z}^{\mathrm{T}} := [z_m^{\mathrm{T}}]_{m=1}^{d(\mathbf{z}^{\mathrm{T}})}$ and visual concepts $\mathbf{z}^{\mathrm{I}} := [z_n^{\mathrm{I}}]_{n=1}^{d(\mathbf{z}^{\mathrm{I}})}$ respectively. The abstract discrete concepts $\mathbf{z}^{\mathrm{T}}$ generate detailed visual concepts $\mathbf{z}^{\mathrm{I}}$ via a sparse graphical model (Figure 1). At a high level, our identification theory consists of two stages. First, we leverage the distribution changes in the vision modality $p(\mathbf{i}|\mathbf{t})$ induced by variations in the text $\mathbf{t}$ to identify atomic vision concepts $z_m^{\mathrm{I}}$. Intuitively, if the visual observation $p(\mathbf{i}|\mathbf{t})$ varies sufficiently, one can discern fundamental visual variations. For example, an infant might learn atomic visual concepts like "fur" and "wings" by observing and comparing grouped images of "dogs" and "birds". As the second step, we show that once visual concepts $z_n^{\mathrm{I}}$ are identified, they can facilitate the identification of atomic text concepts $z_m^{\mathrm{T}}$ provided that the interactions between visual and textual atomic concepts are not overly complex. Returning to the running example, after recognizing atomic concepts "wings" and "fur", the infant can ground these two visual concepts with their corresponding, more abstract textual concepts.

Our theoretical insights lead to a principled T2I generative model **ConceptAligner** in which we explicitly learn

disentangled text and vision representations with sparse connections. In our empirical evaluations, **ConceptAligner** outperforms state-of-the-art text-to-image models in controllable generation tasks. Moreover, thanks to the identifiability theory, **ConceptAligner** offers interpretability, enabling more precise manipulation.

## 2 Related Work

**Concept learning.** A flux of recent work focuses on learning interpretable concepts from images. The concept bottleneck model (Koh et al., 2020) makes predictions on human-annotated concepts and then applies these concepts to downstream tasks. This approach has spurred a significant amount of follow-up work (Zarlenga et al., 2022; Yuksekgonul et al., 2023; Kim et al., 2023; Havasi et al., 2022; Shang et al., 2024; Chauhan et al., 2023). A separate line of research aims to achieve unsupervised concept discovery from vision data by proposing novel neural network architectures and training objectives (Burgess et al., 2019; Locatello et al., 2020; Du et al., 2022a;b; Liu et al., 2023). In contrast, our work considers the theoretical perspective of concept learning and provides reasonable identification conditions with empirical implementations. Rajendran et al. (2024) formulate concepts as affine subspaces of latent variables and provide identifiability guarantees for these subspaces. In contrast, we directly identify each latent variable, which enables us to directly control atomic aspects. Similar to our work, Kong et al. (2024) formulate concept learning as the identification of a discrete hierarchical model and offer theoretical guarantees. However, this work assumes fully discrete latent variables and a single modality, which fails to capture the multimodal alignment problem we consider.

**Causal representation learning.** Causal representation learning seeks to infer high-level causal variables from raw, low-dimensional observations (Schölkopf et al., 2021) with identifiability guarantees. Unfortunately, it has been shown that identifying latent variables in general nonlinear causal models is impossible without additional conditions (Hyvärinen & Pajunen, 1999; Locatello et al., 2019). Existing identifiability conditions include (1) constraints on the generating function (e.g., sparsity) (Xu et al., 2024a; Zheng et al., 2022; Zheng & Zhang, 2023; Buchholz et al., 2022), (2) multiple distributions arising from the same causal model (Hyvarinen et al., 2019; Khemakhem et al., 2020a; Zhang et al., 2024; Kong et al., 2022; von Kügelgen et al., 2024; Jiang & Aragam, 2023; Brehmer et al., 2022; Xie et al., 2023); 3) temporal transitions among latent variables (Yao et al., 2022; 2021; Klindt et al., 2021; Hyvarinen & Morioka, 2017; Lachapelle et al., 2024) and (4) paired multimodal data (e.g., text-image pairs) (Yao et al., 2024; Morioka & Hyvarinen, 2023; 2024; Daunhawer et al., 2023; Sturma et al., 2023; Gresele et al., 2020). Our work aligns most closely with the multimodal category. To contextual-

ize our contributions, Table 1 compares prior multimodal identification theories with ours and we provide detailed discussion in Section 4.

Table 1: **Related work on multimodal causal representation learning.** This table considers whether the latent-variable distribution is nonparametric, whether the identifiability is component-wise, and whether the framework accommodates discrete latent variables.

| Related work | Nonparametric Prior | Component-wise Iden. | Discrete Latents |
| --- | --- | --- | --- |
| von Kügelgen et al. (2021) | ✓ | ✗ | ✗ |
| Daunhawer et al. (2023) | ✓ | ✗ | ✗ |
| Morioka & Hyvarinen (2024) | ✗ | ✓ | ✗ |
| Yao et al. (2024) | ✓ | ✗ | ✗ |
| Sun et al. (2024) | ✓ | ✗ | ✗ |
| **Ours** | ✓ | ✓ | ✓ |

**Controllable text-to-image generation.** ControlGAN (Li et al., 2019) introduces a word-level generator and discriminator to disentangle different visual attributes. Since it is challenging to control the output with a text prompt, many methods resort to using additional supervision, such as canny edges and depth maps (Zhang et al., 2023; Zhao et al., 2024; Mou et al., 2024; Voynov et al., 2023). Some methods also try to inject CLIP image representation to allow variations in the input images (Ye et al., 2023).

**Image editing.** Image editing requires the model to follow semantic instructions to modify an image. One approach is to explore the latent space of the image by the inversion of generative models (Abdal et al., 2019; 2020; Alaluf et al., 2022; Epstein et al., 2022; Xia et al., 2022; Zhu et al., 2020) or by learning an image encoder (Chai et al., 2021; Richardson et al., 2021; Tov et al., 2021), and then edit images through latent vector manipulation. With the advent of CLIP (Radford et al., 2021), which bridges the latent space between images and text, numerous methods (Crowson et al., 2022; Gal et al., 2022b; Kim et al., 2022; Kwon & Ye, 2022; Patashnik et al., 2021; Abdal et al., 2022) have leveraged its capabilities to conduct textual instructions on images. More recently, pre-trained image-text diffusion models, such as Stable Diffusion (Rombach et al., 2022), have further facilitated image editing by providing a robust link between textual instructions and image modifications (Avrahami et al., 2022; Brooks et al., 2023; Ramesh et al., 2022; Hertz et al., 2022; Meng et al., 2021; Kawar et al., 2023). Xu et al. (2024b) focus on designing attention maps to replace the target attention map with the source map. In contrast, we develop superior conditioning representation. These two approaches are complementary.

## 3 Problem Formulation

In this section, we formalize the data-generating process that underlies the interaction between the textual and visual modalities as the foundation for theoretical analysis.

**Notations.** We denote the dimensionality of a multidimensional variable with $d(\cdot)$. We denote the integer set $\{1, \ldots, n\}$ with $[n]$. We refer to a specific coordinate or dimension of a random vector (variable) as a "component", indicated by a subscript.

**Data-generating processes.** We illustrate the data-generating process in Figure 1 and define it as follows.

$$\mathbf{z}^{\mathrm{T}} \sim p(\mathbf{z}^{\mathrm{T}}), \quad \mathbf{z}^{\mathrm{I}} \sim p(\mathbf{z}^{\mathrm{I}} | \mathbf{z}^{\mathrm{T}});$$
$$\mathbf{t} := g^{\mathrm{T}}(\mathbf{z}^{\mathrm{T}}), \quad \mathbf{i} := g^{\mathrm{I}}(\mathbf{z}^{\mathrm{I}}). \tag{1}$$

We denote the observed text as a discrete variable $\mathbf{t} \in \mathcal{T} \subset \mathbb{N}$ and the observed image as a continuous variable $\mathbf{i} \in \mathcal{I} \subset \mathbb{R}^{d(\mathbf{i})}$. [1] The textual representation is a multidimensional discrete variable $\mathbf{z}^{\mathrm{T}} \in \mathcal{Z}^{\mathrm{T}} \subset \mathbb{N}^{d(\mathbf{z}^{\mathrm{T}})}$, which generates the text observation through a mapping $g^{\mathrm{T}} : \mathcal{Z}^{\mathrm{T}} \to \mathcal{T}$. The components $z_m^{\mathrm{T}}$ encode individual atomic textual concepts with potential statistical dependence. Likewise, the visual representation is denoted as $\mathbf{z}^{\mathrm{I}} \in \mathcal{Z}^{\mathrm{I}} \subseteq \mathbb{R}^{d(\mathbf{z}^{\mathrm{I}})}$, which generates the image $\mathbf{i}$ through $g^{\mathrm{I}} : \mathcal{Z}^{\mathrm{I}} \to \mathcal{I}$. Its components $z_n^{\mathrm{I}}$ represent atomic visual concepts in the distribution.

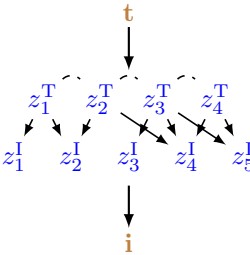

Figure 1: **The data-generating process.** The text-image pair $(\mathbf{t}, \mathbf{i})$ originates from its corresponding representation pair $(\mathbf{z}^{\mathrm{T}}, \mathbf{z}^{\mathrm{I}})$. The textual concepts in $\mathbf{z}^{\mathrm{T}}$ are high-level and govern the generation of relatively low-level visual concepts $\mathbf{z}^{\mathrm{I}}$. The dashed curves indicate potential statistical dependence among the components.

**Atomic visual/textual concepts and their interactions.** We note that "atomic" is defined w.r.t. the distribution. For instance, a component $z_m^{\mathrm{T}}$ in a distribution rich with textual and visual features could encode the concept "dog eyes" whereas in a less diverse distribution, it might encode a coarser concept like "dog faces". The textual representation $\mathbf{z}^{\mathrm{T}}$ serves as an upstream causal variable relative to the visual representation $\mathbf{z}^{\mathrm{I}}$ because text typically contains high-level abstract concepts, while the visual representation includes lower-level details that reflect these concepts. For instance, the visual concept of "dogs" encompasses not only the general idea of a "dog" from the text but also visual details like textures and subtle contours. Importantly, due

---

[1] Note that any multidimensional discrete variable can be converted into a one-dimensional variable via an invertible mapping, so we represent the observed text $\mathbf{t}$ in this way.

to its low-level nature, each visual concept may be influenced by multiple textual concepts. For example, the visual concept "hat" may be affected not only by the textual concept "hat" but also by other textual concepts like "color". These complex relationships are encoded in the graphical model $\mathcal{G}^{t \to i}$. Moreover, statistical dependence may exist among the atomic concepts – for instance, dog species may induce dependence between "eyes" and "noses". Our framework captures these potential dependencies among textual concepts in $\mathbf{z}^T$ (dashed curves in Figure 1).

**Concept identification and alignment for multimodal representations.** Our primary objective is to learn the atomic concepts and their interactions within multimodal data, specifically from textual and visual modalities. By identifying these underlying concepts and modeling their interactions, we *explicitly characterize* how high-level textual concepts influence detailed visual concepts. This foundational characterization enables downstream tasks, such as precise control and manipulation of visual content through textual input. In our formulation, this entails: 1) identifying the atomic textual concepts $z_m^T$ present in the text $\mathbf{t}$; 2) identifying the atomic visual concepts $z_n^I$ in the image $\mathbf{i}$ and 3) learning the interactions between $z_m^T$ and $z_n^I$ through the causal graph $\mathcal{G}^{t \to i}$. By learning these concepts and their interactions, we align the textual and visual representations at an atomic level. This alignment facilitates downstream applications. For example, when editing an image from "a dog" to "a dog with glasses," our model leverages the learned interactions to add the textual concept "glasses," which in turn affects only the corresponding visual concept without inadvertently altering other aspects.

In Section 4, we provide theoretical guarantees and discussions regarding these objectives.

## 4  Identification Theory

In this section, we present the identification conditions and theoretical guarantees that underpin our objectives.

First, we define component-wise identifiability, which serves as a formal definition for disentanglement.

**Definition 4.1** (Component-wise Identifiability)**.** Let $\mathbf{z} \in \mathcal{Z}$ and $\hat{\mathbf{z}} \in \mathcal{Z}$ be variables under two model specifications $\boldsymbol{\theta}$ and $\hat{\boldsymbol{\theta}}$ respectively. We say that $\mathbf{z}$ and $\hat{\mathbf{z}}$ are *identified component-wise* if there exists a permutation $\pi$ such that for each $i \in [d(\mathbf{z})]$, $\hat{z}_i = h_i(z_{\pi(i)})$ where $h_i$ is invertible.

In our context, the two specifications are given by $\boldsymbol{\theta} :=$ $(p(\mathbf{z}^I, \mathbf{z}^T), g^I, g^T)$ and $\hat{\boldsymbol{\theta}} := (p(\hat{\mathbf{z}}^I, \hat{\mathbf{z}}^T), \hat{g}^I, \hat{g}^T)$, where we consider $\boldsymbol{\theta}$ as the true model and $\hat{\boldsymbol{\theta}}$ as its estimate. Under the component-wise identifiability, our estimate $\hat{z}_i$ captures full information of $z_{\pi(i)}$ and no information from $z_j$ such that $j \neq \pi(i)$. This property provides a formal guarantee of disentanglement. The permutation accounts for the

fundamental indeterminacy in the ordering of latent variables (Hyvarinen & Morioka, 2016; Hyvarinen et al., 2019; Kivva et al., 2021).

To achieve our goal, we introduce Condition 4.2 and Condition 4.3, which facilitate the identification of the visual concepts $z_n^I$ and textual concepts $z_m^T$ respectively.

**Condition 4.2** (Visual Concept Identification)**.**

 i *[Invertibility & Smoothness]: Generating functions $g^T$ and $g^I$ are invertible and $g^I$ is smooth.*

 ii *[Smooth and Positive Density]: The probability density function of $\mathbf{z}^I$ is smooth and positive, i.e., $p(\mathbf{z}^I|\mathbf{z}^T) > 0$ is smooth over $\mathcal{Z}^I \times \mathcal{Z}^T$.*

 iii *[Conditional Independence]: Components $z_n^I$ are independent given $\mathbf{z}^T$: $p(\mathbf{z}^I|\mathbf{z}^T) = \prod_n p(z_n^I|\mathbf{z}^T)$.*

 iv *[Sufficient Variability]: For any $\mathbf{z}^I \in \mathcal{Z}^I$, there exist $2d(\mathbf{z}^I) + 1$ values of $\mathbf{z}^T$, i.e., $\mathbf{z}_{(n)}^T$ with $n = 0, 1, \ldots, 2d(\mathbf{z}^I) + 1$, such that the $2d(\mathbf{z}^I)$ vectors $\mathbf{w}(\mathbf{z}^I, \mathbf{z}_{(n)}^T) - \mathbf{w}(\mathbf{z}^I, \mathbf{z}_{(0)}^T)$ are linearly independent, where vector $\mathbf{w}(\mathbf{z}^I, \mathbf{z}^T)$ is defined as follows:*

$$\mathbf{w}(\mathbf{z}_s, \mathbf{z}^T) = \Big( \frac{\partial \log p\left(z_1^I|\mathbf{z}^T\right)}{\partial z_1^I}, \ldots, \frac{\partial \log p\left(z_{d(\mathbf{z}^I)}^I|\mathbf{z}^T\right)}{\partial z_{d(\mathbf{z}^I)}^I},$$
$$\frac{\partial^2 \log p\left(z_1^I|\mathbf{z}^T\right)}{(\partial z_1^I)^2}, \ldots, \frac{\partial^2 \log p\left(z_{d(\mathbf{z}^I)}^I|\mathbf{z}^T\right)}{\partial (z_{d(\mathbf{z}^I)}^I)^2} \Big).$$
$$(2)$$

**Interpretation & discussion.** Condition 4.2 ensures that each visual concept $z_n^I$ can be disentangled from the others. The key idea is that each $z_n^I$ should exhibit sufficiently distinct behavior to be distinguishable. Specifically, Condition 4.2-iv requires the conditional distributions $p(z_m^I|\mathbf{z}^T)$ vary sufficiently over different $\mathbf{z}^T$. Intuitively, when we change the text from "cats" to "dogs", the visual concepts "eyes" and "noses" exhibit different patterns of change, allowing us to recognize them as separate concepts. In general, we expect this condition to hold – standard text-image datasets (e.g., LAION (Schuhmann et al., 2021)) contain millions of captions, far exceeding the number of possible visual concepts. Additionally, we may follow existing methods (e.g., Wang et al. (2023); Chen et al. (2024); Betker et al. (2023)) to employ vision-language models to generate higher-quality captions. Oftentimes, other natural properties can also greatly weaken this condition (e.g., sparsity (Li et al., 2025)). Such sparsity is often encouraged implicitly or explicitly in generative models (e.g., sparse attention patterns). Condition 4.2-i ensures the observed variables preserve all latent variables' information. Otherwise, it would be impossible to recover them from observed variables. These conditions are commonly assumed in the independent component analysis literature (Hyvarinen &

Morioka, 2016; Hyvarinen et al., 2019; Khemakhem et al., 2020a;b; Kong et al., 2022; Zhang et al., 2024).

**Condition 4.3** (Textual Concept Identification)**.**

*i [Non-degeneracy]:* $\mathbb{P}\left[\mathbf{z}^{\mathrm{T}} = k\right]$*, for all* $k \in \Omega$*; for all components* $z^{\mathrm{I}}$*,* $\mathbb{P}\left[z^{\mathrm{I}}|\mathrm{Pa}(z^{\mathrm{I}}) = k_1\right] \neq \mathbb{P}\left[z^{\mathrm{I}}|\mathrm{Pa}(z^{\mathrm{I}}) = k_2\right]$*, if* $k_1 \neq k_2$*.*

*ii [No-twins]: Distinct components* $z^{\mathrm{T}}$ *have distinct neighbors:* $ne(z_m^{\mathrm{T}}) \neq ne(z_n^{\mathrm{T}})$ *for* $m \neq n$*.*

*iii [Maximality]: There is no DAG* $\tilde{\mathcal{G}}^{\mathrm{t}\rightarrow\mathrm{i}}$ *resulting from splitting a latent variable* $z$ *into* $(\tilde{z}_1, \tilde{z}_2)$ *in* $\mathcal{G}^{\mathrm{t}\rightarrow\mathrm{i}}$*, such that the resultant distribution is Markov w.r.t.* $\tilde{\mathcal{G}}^{\mathrm{t}\rightarrow\mathrm{i}}$ *and* $\tilde{\mathcal{G}}^{\mathrm{t}\rightarrow\mathrm{i}}$ *satisfies* ii*.*

*iv [Non-Subset Observed Children]: For any pair* $z_i^{\mathrm{T}}$ *and* $z_j^{\mathrm{T}}$ *with* $i \neq j$*, one's observed children are not the subset of the other's,* $\mathrm{Ch}_{\mathcal{G}^{\mathrm{t}\rightarrow\mathrm{i}}}(z_i^{\mathrm{T}}) \not\subset \mathrm{Ch}_{\mathcal{G}^{\mathrm{t}\rightarrow\mathrm{i}}}(z_j^{\mathrm{T}})$*.*

**Interpretation & discussion.** Condition 4.3 facilitates the identification of discrete textual concepts $z_m^{\mathrm{T}}$. The critical aspect is the sparse connectivity between the visual concepts $z_n^{\mathrm{I}}$ and the textual concepts $z_m^{\mathrm{T}}$, as specified in Condition 4.3-iv. Since the visual concepts $z_n^{\mathrm{I}}$ have already been identified (thanks to Condition 4.2), we can treat them as observed variables. Condition 4.3-iv ensures that each textual concept $z_m^{\mathrm{T}}$ exerts distinguishable influences on the different subsets of visual concepts $z_m^{\mathrm{I}}$, making them identifiable. This condition is reasonable because the textual concepts $z_m^{\mathrm{T}}$ are defined to be atomic and thus should not have heavily overlapping effects on the visual concepts $z_n^{\mathrm{I}}$. Consider concepts like "fur" and "ears" when describing a cat. These concepts should affect partially distinct visual features. If every visual feature triggered by "ears" was also triggered by "fur," these concepts aren't genuinely atomic and should be restructured. Similar conditions have been adopted in prior work (Kivva et al., 2021; 2022; Kong et al., 2024). Condition 4.3-i,ii,iii are a set of necessary conditions for discrete latent variable identification as extensively discussed in prior work (Kivva et al., 2021; 2022). Intuitively, Condition 4.3-i ensures that each discrete latent variable has a detectable influence through the observed variables. Condition 4.3-ii and iii eliminate the indeterminacy arising from arbitrary merging and splitting over latent variables, making identification tractable.

**Theorem 4.4** (Atomic Concept Identification)**.** *We assume the generating process in* (1)*. Under Condition 4.2 and Condition 4.3, we attain component-wise identifiability of concepts* $\mathbf{z}^{\mathrm{T}}$ *and* $\mathbf{z}^{\mathrm{I}}$ *(Definition 4.1) and the bipartite graph* $\mathcal{G}^{\mathrm{t}\rightarrow\mathrm{i}}$ *up to permutation of component indices.*

**Proof sketch.** As we have outlined in discussions on Condition 4.2 and Condition 4.3, we first utilize the variability of

visual concepts $p(z_n^{\mathrm{I}}|\mathbf{t})$ over different textual descriptions $\mathbf{t}$ (since $g^{\mathrm{T}}$ is invertible, varying $\mathbf{t}$ is equivalent to varying $\mathbf{z}^{\mathrm{T}}$) to identify visual concepts $z_n^{\mathrm{I}}$. Subsequently, treating the identified visual concepts $z_n^{\mathrm{I}}$ as observed variables, we exploit the sparse graphical structure between $z_m^{\mathrm{T}}$ and $z_n^{\mathrm{I}}$ (as ensured by Condition 4.3-iv) to disentangle textual concepts $z_m^{\mathrm{T}}$ and recover the causal graph $\mathcal{G}^{\mathrm{t}\rightarrow\mathrm{i}}$.

**Theoretical contribution.** Our work has advanced the theoretical understanding of identifiability in multimodal representation learning. Previous studies (Daunhawer et al., 2023; von Kügelgen et al., 2021; Yao et al., 2024) have established identifiability under shared representation assumptions across modalities. However, their guarantees are limited to identifiability up to subspaces (groups of latent components) determined by the sharing patterns, which is insufficient for tasks requiring precise control over individual concepts. In contrast, we achieve component-wise identifiability of atomic concepts.

Some recent works, such as Morioka & Hyvarinen (2024), also aim for component-wise identifiability but rely on semi-parametric assumptions like the exponential family and additive causal influences. Our fully nonparametric framework offers greater flexibility for modeling complex real-world distributions. While Sun et al. (2024) also leverage sparse connectivity and assume nonparametric models to achieve component-wise identifiability, their approach depends on specific sparsity conditions involving Jacobian matrices of continuous functions, which preclude discrete variables – a capability desired for our problem.

**Practical implications.** As motivated in Section 3, Theorem 4.4 provides the identification guarantee for disentangled atomic textual and visual concepts ($z_m^{\mathrm{T}}$ and $z_n^{\mathrm{I}}$) and their interactions. This foundation is crucial for aligning these two modalities and downstream tasks including T2I generation. In Section 5, we implement the data-generating process and key conditions (e.g., sparse $\mathcal{G}^{\mathrm{t}\rightarrow\mathrm{i}}$ in Condition 4.3-iv) and evaluate our framework in Section 6.

# 5 ConceptAligner : Controllable Text-to-Image with Learning Concepts

Consider a common use case in text-to-image generation: a user provides a text prompt to generate an image, then wishes to make minor edits, such as changing only the color of the clothing. Controllable generation enables the user to modify the prompt accordingly, prompting the model to adjust the specified feature while preserving all other aspects of the image. This ability to make targeted changes without unintended alterations underscores the importance of controllable text-to-image generation.

In this section, we present our empirical approach to controllable text-to-image generation—guided by the identifiability

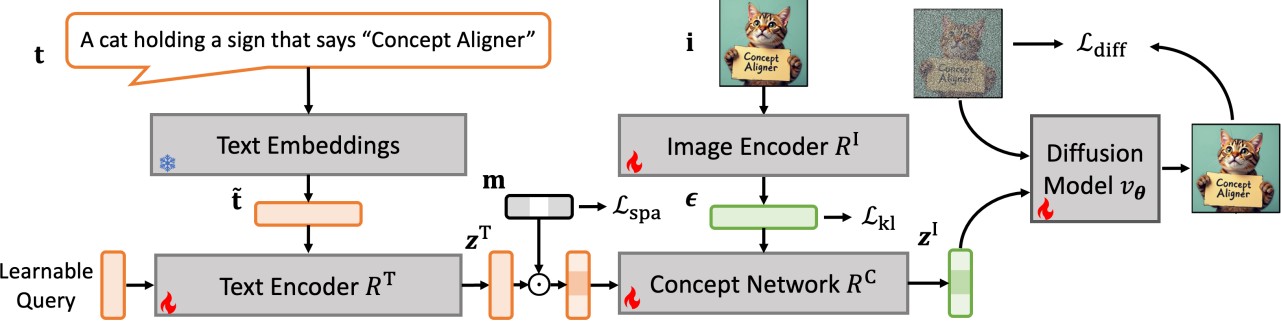

Figure 2: **Diagram of ConceptAligner .** We use text network $R^{\mathrm{T}}$ to recover the textual concepts $\mathbf{z}^{\mathrm{T}}$ and image network $R^{\mathrm{I}}$ to recover the visual exogenous information $\boldsymbol{\epsilon}$. We transform them with our concept network $R^{\mathrm{C}}$ and obtain visual concepts $\mathbf{z}^{\mathrm{I}}$. Finally, we feed the visual concepts $\mathbf{z}^{\mathrm{I}}$ into a diffusion transformer $v_{\boldsymbol{\theta}}$ as the conditioning input.

results established in Section 4—and detail the model architecture, implementation specifics, and training objectives.

## 5.1 Model Design

**Overall structure.** Controllable image generation requires the generative model to faithfully capture the interaction between textual and visual concepts (i.e., $\mathbf{z}^{\mathrm{T}}$ and $\mathbf{z}^{\mathrm{I}}$), aligning the two modalities and enabling precise control from text $\mathbf{t}$ to image $\mathbf{i}$. This alignment allows for precise manipulation of image attributes based on textual conditions. To construct our theoretical framework, our model **ConceptAligner** consists of four major modules:

1. Text network $R^{\mathrm{T}}$ that extracts textual concepts $\mathbf{z}^{\mathrm{T}}$ from the text $\mathbf{t}$;

2. Image network $R^{\mathrm{I}}$ that maps the image $\mathbf{i}$ back to its exogenous information $\boldsymbol{\epsilon}$ associated with the conditional distribution $p(\mathbf{z}^{\mathrm{I}}|\mathbf{z}^{\mathrm{T}})$ (i.e., $\mathbf{z}^{\mathrm{I}} := g_{\mathbf{z}^{\mathrm{I}}}(\mathbf{z}^{\mathrm{T}}, \boldsymbol{\epsilon})$);

3. Concept network $R^{\mathrm{C}}$ that produces the visual concepts $\mathbf{z}^{\mathrm{I}}$ given the textual concepts $\mathbf{z}^{\mathrm{T}}$ and the exogenous information $\boldsymbol{\epsilon}$, i.e., the sampling step $\mathbf{z}^{\mathrm{I}} \sim p(\mathbf{z}^{\mathrm{I}}|\mathbf{z}^{\mathrm{T}})$;

4. Conditional generation model $v_{\boldsymbol{\theta}}$ that renders out the visual representation $\mathbf{z}^{\mathrm{I}}$ to image $\mathbf{i}$.

We present our model **ConceptAligner** in Fig. 2 and introduce each module in detail as follows.

**Text encoder $R^{\mathrm{T}}$.** We first use pre-trained text embeddings to obtain the embedding $\tilde{\mathbf{t}}$ of the text input $\mathbf{t}$. These embeddings retain sequential dependencies and do not represent concepts. To extract meaningful and disentangled concepts from these embeddings, we introduce a perceive-resampler text network $R^{\mathrm{T}}$ (Alayrac et al., 2022), which maps text embeddings $\tilde{\mathbf{t}}$ into a structured concept space (i.e., $\mathbf{z}^{\mathrm{T}}$). We predefine the number of textual concepts and initialize a set

of learnable queries. The final concept $\mathbf{z}^{\mathrm{T}}$ are then obtained by conditioning on the text embedding $\tilde{\mathbf{t}}$ as:

$$\mathbf{z}^{\mathrm{T}} = R^{\mathrm{T}}(\tilde{\mathbf{t}}). \tag{3}$$

**Image encoder $R^{\mathrm{I}}$.** Visual concepts $\mathbf{z}^{\mathrm{I}}$ are sampled from a distribution conditioned on textual concepts $\mathbf{z}^{\mathrm{T}}$, i.e., $p(\mathbf{z}^{\mathrm{I}}|\mathbf{z}^{\mathrm{T}})$. We recover the exogenous information $\boldsymbol{\epsilon}$ involved in the sampling (which can be written as $\mathbf{z}^{\mathrm{I}} := g_{\mathbf{z}^{\mathrm{I}}}(\mathbf{z}^{\mathrm{T}}, \boldsymbol{\epsilon})$) by following the variational autoencoder framework (Kingma, 2013):

$$\begin{aligned} \boldsymbol{\mu}, \sigma &= R^{\mathrm{I}}_{\boldsymbol{\mu}}(\mathbf{i}), R^{\mathrm{I}}_{\sigma}(\mathbf{i}), \\ \boldsymbol{\epsilon} &= \boldsymbol{\mu} + \sigma * \tilde{\boldsymbol{\epsilon}}, \end{aligned} \tag{4}$$

where $R^{\mathrm{I}}$ is the image encoder. We adopt the reparameterization trick and sample $\tilde{\boldsymbol{\epsilon}}$ from the prior $\mathcal{N}(0, \mathbf{I})$.

**Concept network $R^{\mathrm{C}}$.** After obtaining the exogenous variable $\boldsymbol{\epsilon}$ and the textual concepts $\mathbf{z}^{\mathrm{T}}$ in (4) and (3), we implement a concept network $R^{\mathrm{C}}$ to transform them into visual concepts $\mathbf{z}^{\mathrm{I}}$ (i.e., $\mathbf{z}^{\mathrm{I}} \sim p(\mathbf{z}^{\mathrm{I}}|\mathbf{z}^{\mathrm{T}})$):

$$\mathbf{z}^{\mathrm{I}} = R^{\mathrm{C}}\left(\mathbf{z}^{\mathrm{T}} \odot \mathbf{m}, \boldsymbol{\epsilon}\right), \tag{5}$$

where $\mathbf{m}$ is a learnable mask whose components take values in $[0, 1]$ to control the sparsity of the connectivity from $\mathbf{z}^{\mathrm{T}}$ to $\mathbf{z}^{\mathrm{I}}$ (i.e., $\mathcal{G}^{\mathrm{t}\rightarrow\mathrm{i}}) - m_{i,j} = 0$ indicates no influence from the textual concepts $z_i^{\mathrm{T}}$ to the visual concepts $z_j^{\mathrm{I}}$.

**Conditional generation $v_{\boldsymbol{\theta}}$.** We utilize a diffusion transformer $v_{\boldsymbol{\theta}}$ for our conditional generation. It takes as input a noisy image and certain conditioning to perform denoising. In the conventional T2I model, the conditioning is the text embedding from a pre-trained text encoder (e.g., CLIP (Radford et al., 2021)). In our method, we use the visual representation $\mathbf{z}^{\mathrm{I}}$ as the conditioning.

## 5.2 Loss Functions

As motivated before, our empirical goals include

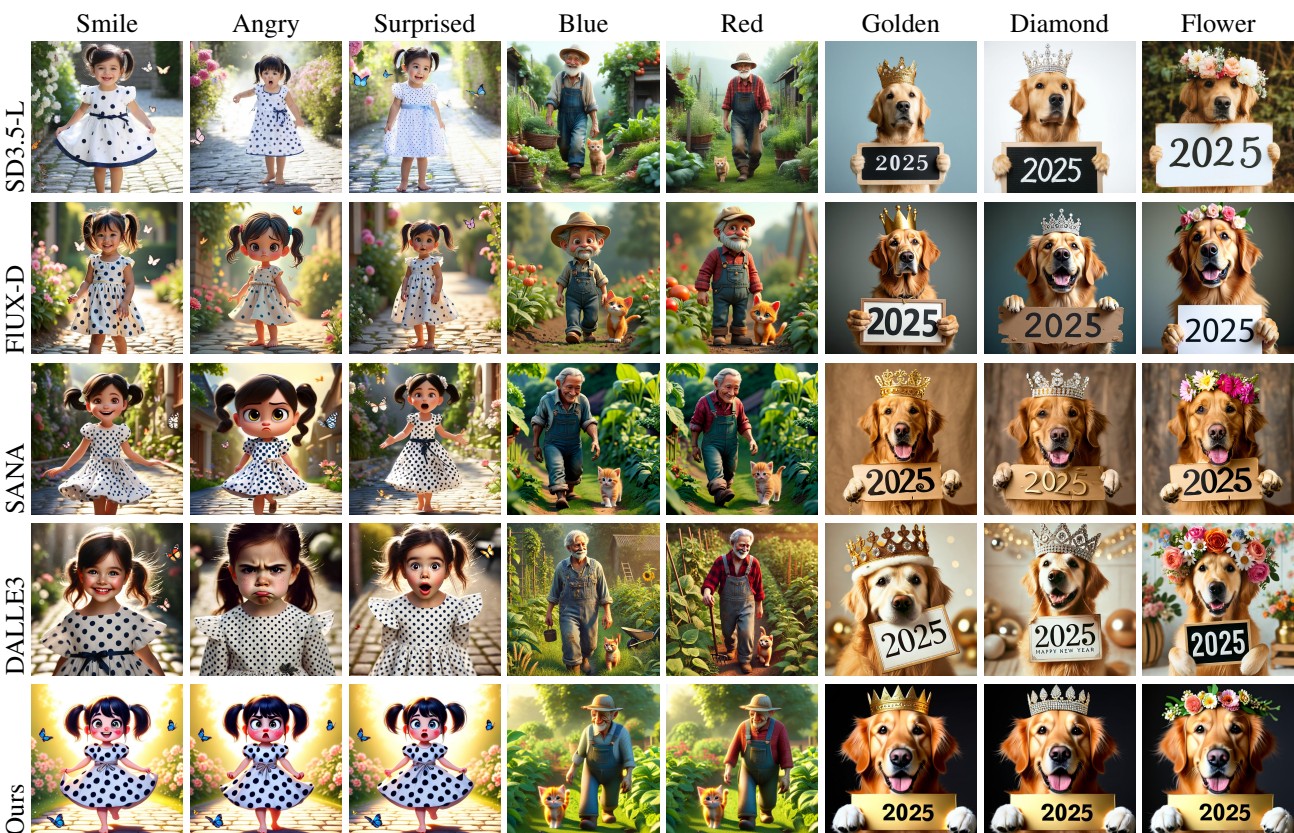

Figure 3: **Controllable text-to-image generation comparisons.** Thanks to the identifiability of visual and textual concepts, our method can make only necessary changes without affecting other attributes. For example, when we modify the prompt from "golden crown" to "flower crown" for the dog, SD-3.5-L changes the color of the sign from black to white background whereas our model precisely implements the requested change.

1. T2I generation: accurately generating images $\mathbf{i}$ based on the textual description $\mathbf{t}$.

2. Text-based editing: applying changes specified in a new textual instruction $\tilde{\mathbf{t}}$ to a previously generated image $\mathbf{i}$, without introducing unnecessary modifications.

Having introduced the architecture of our model, we now present the training losses to achieve the above goals.

**Diffusion loss.** As mentioned previously, we condition the diffusion transformer model with our visual concepts $\mathbf{z}^{\mathrm{I}}$. We set the initial diffusion sample $\mathbf{i}_0 := \mathbf{i}$ and draw random noises $\boldsymbol{\eta}$ to corrupt the image $\mathbf{i}_0$. We adopt the denoising loss (Liu et al., 2022; Lipman et al., 2022; Xie et al., 2024; Esser et al., 2024),

$$\mathbf{i}_\alpha = (1-\alpha)\mathbf{i}_0 + \alpha\boldsymbol{\eta}, \tag{6}$$
$$\mathcal{L}_{\mathrm{diff}} = \mathbb{E}_{\boldsymbol{\eta}\sim\mathcal{N}(0,\mathbf{I})}\|v_{\boldsymbol{\theta}}(\mathbf{i}_\alpha, \alpha, \mathbf{z}^{\mathrm{I}}) - (\boldsymbol{\eta} - \mathbf{i}_0)\|^2,$$

where $\alpha \in (0,1)$ is sampled from a predefined distribution, e.g., uniform distribution $\mathcal{U}[0,1]$.

**Kullback–Leibler divergence loss.** We apply an image encoder to recover the exogenous information $\boldsymbol{\epsilon}$, which is sampled from a prior distribution $\mathcal{N}(0,\mathbf{I})$. To enforce this, we match the marginal distribution of $\boldsymbol{\epsilon}$ with the prior distribution as

$$\mathcal{L}_{\mathrm{kl}} = -\log(\sigma) + \frac{\sigma^2 + \boldsymbol{\mu}^2}{2} - \frac{1}{2}. \tag{7}$$

**Sparsity regularization.** As indicated in Condition 4.3-iv, the connections from $\mathbf{z}^{\mathrm{T}}$ to $\mathbf{z}^{\mathrm{I}}$ should be sparse enough to recover the textual concepts $z_m^{\mathrm{T}}$. In our implementation, we use a learnable mask $\mathbf{m}$ to modulate the connectivity by applying a sparsity regularization:

$$\mathcal{L}_{\mathrm{spa}} = \|\mathbf{m}\|_1. \tag{8}$$

**Full objective.** Combining the above loss functions, we arrive at our final objective:

$$\mathcal{L} = \mathcal{L}_{\mathrm{diff}} + \lambda_{\mathrm{spa}}\mathcal{L}_{\mathrm{spa}} + \lambda_{\mathrm{kl}}\mathcal{L}_{\mathrm{kl}}, \tag{9}$$

where $\lambda_{\mathrm{spa}}$ and $\lambda_{\mathrm{kl}}$ balance the three loss terms.

| Method | CLIP-I ↑ | LPIPS ↓ | CLIP-T ↑ | DINO ↑ |
|---|---|---|---|---|
| SD3.5-M | 0.862 | 0.428 | **0.321** | 0.719 |
| SD3.5-L | 0.864 | 0.456 | 0.318 | 0.700 |
| FLUX-S | 0.868 | 0.463 | 0.318 | 0.740 |
| FLUX-D | 0.872 | 0.452 | 0.310 | 0.721 |
| SANA | 0.870 | 0.438 | 0.313 | 0.741 |
| SANA-Finetune | 0.865 | 0.457 | 0.320 | 0.750 |
| **ConceptAligner** | **0.903** | **0.357** | 0.314 | **0.835** |
| w.o sparsity | 0.863 | 0.388 | 0.308 | 0.751 |
| w.o diffusion | 0.999 | 0.000 | 0.171 | 0.999 |
| w.o KL | 0.909 | 0.246 | 0.206 | 0.826 |

Table 2: **Comparisons on Emu-Edit dataset.** We generate pairs of images with the source and target prompt in Emu-edit dataset (Sheynin et al., 2024). Our method achieves the best or competitive performance across various metrics.

# 6 Experiments

In this section, we first describe the experimental setup, including implementation details, datasets, baselines, and evaluation metrics. We then present the results, covering comparisons with baselines, visualizations of learned concepts, disentanglement analysis, and ablation studies. More results and analyses can be located in Appendix B.

## 6.1 Setup

**Implementation.** We implement our method based on SANA (Xie et al., 2024). Firstly, we employ a 6-block perceiver resampler (Alayrac et al., 2022) to transform the entangled text embedding into our textual concept $\mathbf{z}^{\mathrm{T}}$. We define the number of textual tokens as 64. Then we use a transformer block to transform Siglip (Zhai et al., 2023) image embedding into the mean and variance of the latent $\epsilon$. Then we feed the re-parametrized latent into a 6-block perceiver resampler with masking $\mathbf{m} \odot \mathbf{z}^{\mathrm{T}}$. Finally, we obtain the image representation $\mathbf{z}^{\mathrm{I}}$ and replace the original text embedding with this representation. We use LoRA on the diffusion transformer with rank 256. All the parameters are trained with batch size 768 and learning rate $5 \cdot 10^{-5}$.

**Datasets.** We use FLUX-S (Labs, 2024) to generate 2 million images using prompts sourced from the LAION dataset. Subsequently, we employ QWEN2-VL (Wang et al., 2024b) to produce accurate textual descriptions.

To evaluate the controllability of our generative model, we need to generate pairs of images that reflect specific target changes. For this purpose, we utilize the EMU-Edit dataset (Sheynin et al., 2024), which includes 3,589 paired prompts spanning seven image editing categories. Using the provided source and target prompts, we generate corresponding image pairs for analysis.

**Baselines.** We compare with state-of-the-art methods for

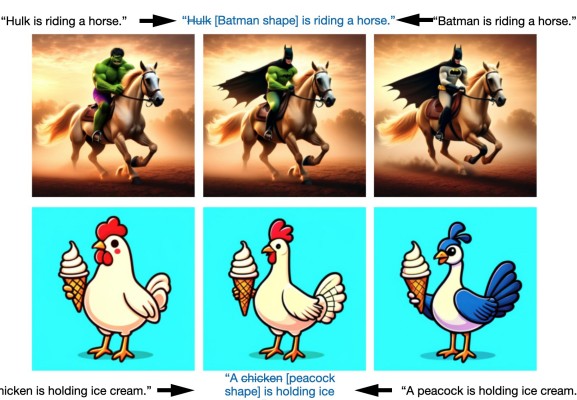

Figure 4: **Visualization of the learned concepts.** From the text embeddings of "Batman" and "peacock", we isolate distinct visual elements - specifically, the flowing cape characteristic of Batman and the distinctive shape associated with peacocks. Our approach enables selective modification of individual features while preserving all other aspects. This is demonstrated in the middle columns, where we can substitute one learned feature from the left column with a corresponding feature from the right column, resulting in coherent images that differ only in that specific aspect.

T2I generation. Specifically, we compare with stable-diffusion3.5-medium (SD3.5-M) (Esser et al., 2024), stable-diffusion3.5-large (SD3.5-L) (Esser et al., 2024), Flux.1-dev (FLUX-D) (Labs, 2024), Flux.1-Schnell (FLUX-S) (Labs, 2024) and SANA (Xie et al., 2024). For a fair comparison, we also finetune SANA on our training data and refer to the resulting model as SANA-Finetune. We fix the random seed for each pair of generations.

**Metrics.** To evaluate the similarity between the generated pairs of images, we calculate the CLIP (Radford et al., 2021) and DINO (Caron et al., 2021) embedding similarities, along with the LPIPS (Zhang et al., 2018) score. Additionally, we assess the average CLIP image-text similarity to determine how well the generated image aligns with the given prompts.

## 6.2 Results

**Comparisons with baselines.** We present our text-to-image generation results in Table 2. As shown, **ConceptAligner** outperforms other methods in terms of CLIP-I, DINO similarity, and LPIPS scores. Figure 3 further illustrates that our model can apply the intended changes while preserving unrelated attributes—for example, maintaining the color of the sign or the pose of the dog.

**Analysis of the concepts.** To analyze the learned concepts, we interpolate between them to generate new images and compare these with the originals. As illustrated in Fig. 4, **ConceptAligner** effectively disentangles complex text em-

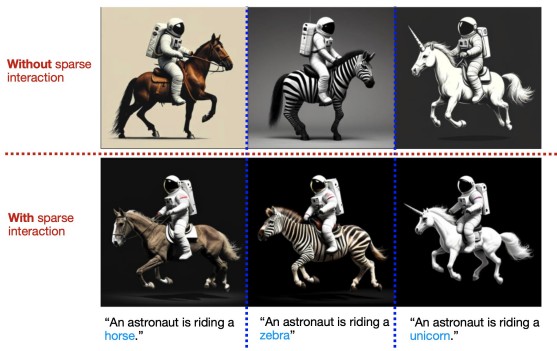

"An astronaut is riding a horse."  "An astronaut is riding a zebra"  "An astronaut is riding a unicorn."

Figure 5: **Comparisons with the model trained without sparsity regularization.** Without applying sparsity regularization, the model makes unnecessary changes when we alter the text input from "horse" to "zebra".

| Method | Subject ↑ | Prompt ↑ |
|---|---|---|
| SANA (Xie et al., 2024) | 0.913 | 0.615 |
| SANA-Finetune (Xie et al., 2024) | 0.934 | 0.731 |
| **ConceptAligner** | **0.946** | **0.814** |

Table 3: **Disentanglement comparisons.** We modify the subject's action in the source prompt and evaluate whether the generative models can preserve the subject's identity, given that only the action in the text input is altered. Our method achieves the highest scores in both subject consistency and prompt consistency.

beddings into atomic textual concepts. For instance, in the second row, we recover the shape concept of a peacock and successfully generate a chicken with a peacock shape. These results demonstrate that our method learns to decompose text embeddings into interpretable, atomic concepts.

**Disentanglement analysis**. If the model learns disentangled representations, it should be able to preserve unrelated concepts when only specific parts of the text are modified. To evaluate this, we prompt ChatGPT to randomly generate 10 animal names and 10 actions. For each animal, we fix its identity and modify only the action, resulting in 100 original–edited image pairs. This process is repeated across 10 random seeds, yielding a total of 1,000 pairs.

For evaluation, we use Qwen2.5-VL-Instruct-7B to assess two criteria: subject consistency, measuring whether the animal identity is preserved, and prompt consistency, evaluating whether the intended action change is accurately reflected. **ConceptAligner** achieves the highest scores on both metrics compared to SANA and finetuned SANA, demonstrating superior disentanglement capabilities.

**Ablation study.** We present the quantitative ablation results in Table 2. As outlined in our theoretical framework, achieving sparse connections from textual concepts $\mathbf{z}^T$ to visual

concepts $\mathbf{z}^I$ is crucial, which motivates the need for sparsity regularization. To assess this, we train a model without the masking module, resulting in a fully connected $\mathbf{z}^T \to \mathbf{z}^I$ mapping. A qualitative comparison of outputs from both models is shown in Fig. 5. We observe that the model lacking sparsity introduces more irrelevant changes when the prompt is modified from horse to zebra, underscoring the importance of promoting sparse $\mathbf{z}^T \to \mathbf{z}^I$ connections, consistent with our identifiability theory.

Additionally, removing the diffusion loss results in nearly identical image pairs, indicating that the model fails to leverage the paired prompts. Eliminating the KL divergence loss causes the exogenous variable $\epsilon$ to carry excessive information about the clean image, allowing the model to ignore the text condition for denoising, and ultimately fail to produce prompt-consistent outputs.

# 7 Conclusion

In this work, we develop theoretical foundations for learning atomic concepts and their interactions from multimodal data distributions. In comparison with prior theoretical work Yao et al. (2024); von Kügelgen et al. (2021); Daunhawer et al. (2023); Gresele et al. (2020); Sun et al. (2024); Morioka & Hyvarinen (2024; 2023), our theory provides component-wise identifiability for each atomic concept without resorting to semi-parametric assumptions, while admitting modalities consisting of continuous and discrete latent variables. As a consequence of our theory, we design a principled T2I generative model. Under thorough evaluation, our model outperforms baselines in T2I generation tasks and demonstrates superior interpretability and controllability. **Limitations:** in order to identify visual atomic concepts $z_n^I$, Condition 4.2-iv demands the text $\mathbf{t}$ to impose sufficient influences over the visual concept distribution $p(\mathbf{z}^I|\mathbf{t})$. Practically, this requires captions in the dataset to be sufficiently informative and diverse, which may be violated on datasets with poor caption qualities.

## Impact Statement

This paper presents work whose goal is to advance the field of Machine Learning. There are many potential societal consequences of our work, none which we feel must be specifically highlighted here.

## Acknowledgments

We appreciate the anonymous reviewers and the area chair for their insightful feedback. We would also like to acknowledge the support from NSF Award No. 2229881, AI Institute for Societal Decision Making (AI-SDM), the National Institutes of Health (NIH) under Contract R01HL159805, and grants from Quris AI, Florin Court Capital, and MBZUAI-WIS Joint Program. The work of L. Kong is supported in part by NSF DMS-2134080 through an award to Y. Chi.

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

# A  Proof for Theorem 4.4

We present the proof for Theorem 4.4. We duplicate the theorem here for readability.

**Condition 4.2** (Visual Concept Identification)**.**

   i *[Invertibility & Smoothness]: Generating functions $g^{\mathrm{T}}$ and $g^{\mathrm{I}}$ are invertible and $g^{\mathrm{I}}$ is smooth.*

   ii *[Smooth and Positive Density]: The probability density function of $\mathbf{z}^{\mathrm{I}}$ is smooth and positive, i.e., $p(\mathbf{z}^{\mathrm{I}}|\mathbf{z}^{\mathrm{T}}) > 0$ is smooth over $\mathcal{Z}^{\mathrm{I}} \times \mathcal{Z}^{\mathrm{T}}$.*

   iii *[Conditional Independence]: Components $z_n^{\mathrm{I}}$ are independent given $\mathbf{z}^{\mathrm{T}}$: $p(\mathbf{z}^{\mathrm{I}}|\mathbf{z}^{\mathrm{T}}) = \prod_n p(z_n^{\mathrm{I}}|\mathbf{z}^{\mathrm{T}})$.*

   iv *[Sufficient Variability]: For any $\mathbf{z}^{\mathrm{I}} \in \mathcal{Z}^{\mathrm{I}}$, there exist $2d(\mathbf{z}^{\mathrm{I}}) + 1$ values of $\mathbf{z}^{\mathrm{T}}$, i.e., $\mathbf{z}_{(n)}^{\mathrm{T}}$ with $n = 0, 1, \ldots, 2d(\mathbf{z}^{\mathrm{I}}) + 1$, such that the $2d(\mathbf{z}^{\mathrm{I}})$ vectors $\mathbf{w}(\mathbf{z}^{\mathrm{I}}, \mathbf{z}_{(n)}^{\mathrm{T}}) - \mathbf{w}(\mathbf{z}^{\mathrm{I}}, \mathbf{z}_{(0)}^{\mathrm{T}})$ are linearly independent, where vector $\mathbf{w}(\mathbf{z}^{\mathrm{I}}, \mathbf{z}^{\mathrm{T}})$ is defined as follows:*

$$\mathbf{w}(\mathbf{z}_s, \mathbf{z}^{\mathrm{T}}) = \left( \frac{\partial \log p\left(z_1^{\mathrm{I}}|\mathbf{z}^{\mathrm{T}}\right)}{\partial z_1^{\mathrm{I}}}, \ldots, \frac{\partial \log p\left(z_{d(\mathbf{z}^{\mathrm{I}})}^{\mathrm{I}}|\mathbf{z}^{\mathrm{T}}\right)}{\partial z_{d(\mathbf{z}^{\mathrm{I}})}^{\mathrm{I}}}, \frac{\partial^2 \log p\left(z_1^{\mathrm{I}}|\mathbf{z}^{\mathrm{T}}\right)}{(\partial z_1^{\mathrm{I}})^2}, \ldots, \frac{\partial^2 \log p\left(z_{d(\mathbf{z}^{\mathrm{I}})}^{\mathrm{I}}|\mathbf{z}^{\mathrm{T}}\right)}{\partial (z_{d(\mathbf{z}^{\mathrm{I}})}^{\mathrm{I}})^2} \right) \tag{2}$$

**Condition 4.3** (Textual Concept Identification)**.**

   i *[Non-degeneracy]: $\mathbb{P}\left[\mathbf{z}^{\mathrm{T}} = k\right]$, for all $k \in \Omega$; for all components $z^{\mathrm{I}}$, $\mathbb{P}\left[z^{\mathrm{I}}|\mathrm{Pa}(z^{\mathrm{I}}) = k_1\right] \neq \mathbb{P}\left[z^{\mathrm{I}}|\mathrm{Pa}(z^{\mathrm{I}}) = k_2\right]$, if $k_1 \neq k_2$.*

   ii *[No-twins]: Distinct components $z^{\mathrm{T}}$ have distinct neighbors: $ne(z_m^{\mathrm{T}}) \neq ne(z_n^{\mathrm{T}})$ for $m \neq n$.*

   iii *[Maximality]: There is no DAG $\tilde{\mathcal{G}}^{\mathrm{t} \to \mathrm{i}}$ resulting from splitting a latent variable $z$ into $(\tilde{z}_1, \tilde{z}_2)$ in $\mathcal{G}^{\mathrm{t} \to \mathrm{i}}$, such that the resultant distribution is Markov w.r.t. $\tilde{\mathcal{G}}^{\mathrm{t} \to \mathrm{i}}$ and $\tilde{\mathcal{G}}^{\mathrm{t} \to \mathrm{i}}$ satisfies ii.*

   iv *[Non-Subset Observed Children]: For any pair $z_i^{\mathrm{T}}$ and $z_j^{\mathrm{T}}$ with $i \neq j$, one's observed children are not the subset of the other's, $\mathrm{Ch}_{\mathcal{G}^{\mathrm{t} \to \mathrm{i}}}(z_i^{\mathrm{T}}) \not\subset \mathrm{Ch}_{\mathcal{G}^{\mathrm{t} \to \mathrm{i}}}(z_j^{\mathrm{T}})$.*

**Theorem 4.4** (Atomic Concept Identification)**.** *We assume the generating process in* (1)*. Under Condition 4.2 and Condition 4.3, we attain component-wise identifiability of concepts $\mathbf{z}^{\mathrm{T}}$ and $\mathbf{z}^{\mathrm{I}}$ (Definition 4.1) and the bipartite graph $\mathcal{G}^{\mathrm{t} \to \mathrm{i}}$ up to permutation of component indices.*

*Proof.* The proof consists of two main steps. In **Step 1**, we take advantage of the observed discrete variable $\mathbf{t}$ and its influence over the visual latent variable $\mathbf{z}^{\mathrm{I}}$ (Condition 4.2-iv) to establish the component-wise identifiability of $\mathbf{z}^{\mathrm{I}}$. This proof technique is adopted in the causal representation learning and nonlinear independent component analysis community (Hyvarinen et al., 2019; Khemakhem et al., 2020a; Kong et al., 2022).

In **Step 2**, we leverage the identified visual latent components $\mathbf{z}^{\mathrm{I}}$ to further identify the textual latent variables $\mathbf{z}^{\mathrm{T}}$, together with the graph $\mathcal{G}^{\mathrm{t} \to \mathrm{i}}$ that connects the textual representation (atomic concepts) to the visual representation (atomic concepts). The crux of this step is to treat the $\mathbf{z}^{\mathrm{I}}$ components as observed variables since they can be identified from the image data $\mathbf{i}$, as shown in **Step 1**, and also utilize the sparsity constraint on the graph $\mathcal{G}^{\mathrm{t} \to \mathrm{i}}$ (Condition 4.3-iv).

**Step 1.** We first introduce Lemma A.1, which we adapt from Kong et al. (2022) by omitting their invariant latent subspace.

**Lemma A.1** (Adapted from Kong et al. (2022))**.** *We follow the following data-generation process*

$$\mathbf{z} \sim p(\mathbf{z}|\mathbf{u}), \ \mathbf{x} := g(\mathbf{z}) \tag{10}$$

*and make the following assumptions:*

   • *A1 (Smooth and Positive Density): The probability density function of latent variables is smooth and positive, i.e., $p_{\mathbf{z}|\mathbf{u}}$ is smooth and $p_{\mathbf{z}|\mathbf{u}} > 0$ over $\mathcal{Z}$ and $\mathcal{U}$.*

   • *A2 (Conditional independence): Conditioned on $\mathbf{u}$, each $z_i$ is independent of any other $z_j$ for $i, j \in [n]$, $i \neq j$, i.e. $\log p_{\mathbf{z}|\mathbf{u}}(\mathbf{z}|\mathbf{u}) = \sum_i^n q_i(z_i, \mathbf{u})$ where $q_i$ is the log density of the conditional distribution, i.e., $q_i := \log p_{z_i|\mathbf{u}}$.*

- *A3 (Linear independence): For any $\mathbf{z} \in \mathcal{Z}$, there exist $2d(\mathbf{z}) + 1$ values of $\mathbf{u}$, i.e., $\mathbf{u}_j$ with $j = 0, 1, \ldots, 2d(\mathbf{z})$, such that the $2n$ vectors $\mathbf{w}(\mathbf{z}, \mathbf{u}_j) - \mathbf{w}(\mathbf{z}, \mathbf{u}_0)$ are linearly independent, where vector $\mathbf{w}(\mathbf{z}, \mathbf{u})$ is defined as follows:*

$$\mathbf{w}(\mathbf{z}_s, \mathbf{u}) = \Big( \frac{\partial \log p(z_1|\mathbf{u})}{\partial z_1}, \ldots, \frac{\partial \log p(z_{d(\mathbf{z})}|\mathbf{u})}{\partial z_n}, \frac{\partial^2 \log p(z_1|\mathbf{u})}{\partial z_1^2}, \ldots, \frac{\partial^2 \log p(z_{d(\mathbf{z})}|\mathbf{u})}{\partial z_{d(\mathbf{z})}^2} \Big). \tag{11}$$

*We can achieve component-wise identifiability of the latent variable $\mathbf{z}$.*

Lemma A.1 leverages the conditional distribution index $\mathbf{u}$ for identification. Since we have access to text $\mathbf{t}$ which is equivalent to the textual concept variable $\mathbf{z}^{\mathrm{T}}$ up to an invertible map, we can simplify the data-generating process in (1) by omitting $\mathbf{z}^{\mathrm{T}}$:

$$\mathbf{t} \sim p(\mathbf{t}), \ \mathbf{z}^{\mathrm{I}} \sim p(\mathbf{z}^{\mathrm{I}}|\mathbf{t}), \ \mathbf{i} := g^{\mathrm{I}}(\mathbf{z}^{\mathrm{I}}). \tag{12}$$

We observe that this data-generating process is a special case of Lemma A.1 where we treat the text variable $\mathbf{t}$ as the distribution index $\mathbf{u}$ in Lemma A.1. As a consequence, under Condition 4.2, we can show visual atomic concepts $z_n^{\mathrm{I}}$ are identifiable, thanks to the distribution variability (i.e., the changes of the conditional distribution $p(\mathbf{z}^{\mathrm{I}}|\mathbf{z}^{\mathrm{T}})$) resulting from the discrete variable $\mathbf{t}$.

**Step 2.** Given the identified visual atomic concepts $z_n^{\mathrm{I}}$, we can simplify our data-generating process as:

$$\mathbf{z}^{\mathrm{T}} \sim p(\mathbf{z}^{\mathrm{T}}), \ \mathbf{z}^{\mathrm{I}} \sim p(\mathbf{z}^{\mathrm{I}}|\mathbf{z}^{\mathrm{T}}), \tag{13}$$

where we treat the identified visual concepts $z_n^{\mathrm{I}}$ as observed variables and view textual concepts $z_m^{\mathrm{T}}$ as latent discrete variables with potential statistical dependence. These two sets of variables are connected via a bipartite graph $\mathcal{G}^{\mathrm{t} \to \mathrm{i}}$, which we assume to be sparse (Condition 4.3-iv).

In the following, we present a lemma adapted from Kivva et al. (2021) that utilizes the sparse graphical structure of $\mathcal{G}^{\mathrm{t} \to \mathrm{i}}$ to component-wise identify the textual atomic concepts $z_m^{\mathrm{T}}$.

**Definition A.2** (Mixture Oracles (Kivva et al., 2021)). Let $\mathbf{z}^{\mathrm{I}}$ be a set of observed variables and $\mathbf{t} \in \Omega \subset \mathbb{N}$ be the discrete variable that is an invertible function of the latent discrete variable $\mathbf{z}^{\mathrm{T}} \in \mathbb{N}^{d(\mathbf{z}^{\mathrm{T}})}$. The mixture model is defined as $\mathbb{P}[\mathbf{z}^{\mathrm{I}}] = \sum_{k \in \Omega} \mathbb{P}[\mathbf{t} = k] \mathbb{P}[\mathbf{z}^{\mathrm{I}}|\mathbf{t} = k]$. A mixture oracle $\mathrm{MixOracle}(\mathbf{z}^{\mathrm{I}})$ takes $\mathbb{P}[\mathbf{z}^{\mathrm{I}}]$ as input and returns the number of components $|\Omega|$, the weights $\mathbb{P}[\mathbf{t} = k]$ and the component $\mathbb{P}[\mathbf{z}^{\mathrm{I}}|\mathbf{t} = k]$ for $k \in \Omega$.

**Lemma A.3** (adapted from Kivva et al. (2021)). *We assume the generating process (13). Under Condition 4.3, one can reconstruct the bipartite graph $\mathcal{G}^{\mathrm{t} \to \mathrm{i}}$ between $\mathbf{z}^{\mathrm{I}}$ and $\mathbf{z}^{\mathrm{T}}$, the invertible map $g^{\mathrm{T}} : \mathbf{z}^{\mathrm{T}} \mapsto \mathbf{t}$, and the joint distribution $\mathbb{P}\left[z_1^{\mathrm{T}} = k_1, \ldots, z_{d(\mathbf{z}^{\mathrm{T}})}^{\mathrm{T}} = k_{d(\mathbf{z}^{\mathrm{T}})}\right]$ from $\mathbb{P}[\mathbf{z}^{\mathrm{I}}]$ and $\mathrm{MixOracle}(\mathbf{z}^{\mathrm{I}})$.*

We note that since we have access to the joint distribution $p(\mathbf{t}, \mathbf{i})$ and have identified $\mathbf{z}^{\mathrm{I}}$ component-wise, we can derive the $\mathrm{MixOracle}(\mathbf{z}^{\mathrm{I}})$ from $p(\mathbf{t}, \mathbf{z}^{\mathrm{I}})$ by marginalizing out components in $\mathbf{z}^{\mathrm{I}}$ accordingly. Given this equivalent formulation, we invoke Lemma A.3 to establish the identifiability of the map $g^{\mathrm{T}}$, from which we can component-wise identify $z_m^{\mathrm{T}}$ from the observed variable $\mathbf{t}$.

$\square$

# B More Empirical Results

**Generation results on PIE-BENCH dataset Ju et al. (2024).** We also evaluate our method on the PIE-BENCH dataset (Ju et al., 2024). As shown in Table 4, our method achieves superior or competitive performance across various metrics. In order to test the model's capability when given long captions, we apply QWEN2.5-Instruct-32B to expand the prompts in PIE-BENCH. The average caption length increases from 8 to 68. As we can see from Table 5, our method achieves better results than the strong baselines SANA and SANA-Fintune.

**Generation samples for EMU-Edit dataset (Sheynin et al., 2024).** In the main paper, we present quantitative results on the EMU-Edit dataset. As shown in Fig. 7, our method produces more consistent paired images compared to the baselines FLUX-D and SANA.

**Sampling efficiency.** Our method builds upon SANA (Xie et al., 2024) and introduces three lightweight networks during training. However, only the textual and concept networks are used during inference for text-to-image generation. One potential concern is the additional computational cost introduced by these new components. To address this, we present a comparison of sampling efficiency in Table 6. Despite the added networks, our method is the fastest in generating an image. This efficiency is primarily due to our use of a more compact representation, whereas SANA relies on 300 text tokens during inference.

**Ablation samples.** In the main paper, we present quantitative comparisons by systematically removing each proposed module. Fig. 6 shows qualitative examples from different model variants. When sparsity regularization is omitted, the generated images exhibit greater inconsistency, indicating that dense connections between textual and visual concepts hinder controllability. Removing the diffusion loss, which serves as the primary objective for training the diffusion model, leads to outputs that lack meaningful structure. Additionally, excluding the KL loss, which regularizes the information contained in the exogenous variable $\epsilon$, results in $\epsilon$ encoding excessive details about the input clean image. Consequently, the model ignores the text prompt for denoising, producing images that fail to reflect the input prompt accurately.

**Multiple-concepts editing.** It is also interesting to investigate whether our model supports multi-concept editing. To explore this, we present examples in Fig. 8, showing results where 2, 3, or 4 concepts are edited simultaneously. As observed, the baseline SANA often alters the subject's identity or the overall image layout, even when such changes are not reflected in the modified prompt. In contrast, our method produces paired images that accurately reflect the intended edits while preserving unrelated attributes, demonstrating stronger control and disentanglement.

**Disentanglement samples.** In Fig. 9, we present paired samples generated for disentanglement analysis, where only the action or background in the text prompt is modified. We observe that both SANA and SANA-Finetune frequently alter the subject's identity, despite changes in the text inputs being limited to action or background. In contrast, our method more effectively preserves the animal's identity, demonstrating stronger disentanglement of concepts.

**Real-world image editing.** While the primary focus of our work is controllable text-to-image generation, it is also important to evaluate the model's ability to handle real-world image editing tasks. To this end, we conduct experiments on a subset of the PIE-Bench dataset. The results, shown in Table 7, demonstrate that our method achieves the best or competitive performance across various metrics. Furthermore, as illustrated in Fig. 10, our method applies precise and necessary edits to the input image, avoiding excessive or unintended modifications that can lead to visual distortions.

| Method | CLIP-I ↑ | LPIPS ↓ | CLIP-T↑ | DINO ↑ |
|---|---|---|---|---|
| SD3.5-M | 0.822 | 0.533 | 0.339 | 0.565 |
| SD3.5-L | 0.824 | 0.544 | **0.340** | 0.516 |
| FLUX-S | 0.837 | 0.536 | 0.331 | 0.603 |
| FLUX-D | 0.823 | 0.566 | 0.328 | 0.505 |
| SANA | 0.839 | 0.573 | 0.331 | 0.593 |
| SANA-Finetune | 0.840 | 0.574 | 0.334 | 0.603 |
| **ConceptAligner** | **0.879** | **0.488** | 0.331 | **0.651** |

Table 4: **Evaluation results on controllable text-to-image generation on PIE-Bench dataset.** We add SANA-Fintune as an additional baseline method where we apply LoRA to fine-tune SANA on our training dataset for comparison. We also include DINO similarity as one metric. We can observe that **ConceptAligner** consistently outperforms or is comparable to the baselines across all metrics. We bold the best performances.

| Method | CLIP-I ↑ | LPIPS ↓ | CLIP-T ↑ | DINO ↑ |
|---|---|---|---|---|
| SANA | 0.788 | 0.680 | 0.328 | 0.406 |
| SANA-Finetune | 0.786 | 0.707 | **0.329** | 0.403 |
| **ConceptAligner** | **0.826** | **0.596** | 0.328 | **0.549** |

Table 5: **Evaluation results on long captions**. We use QWEN2.5-Instruct-32B to expand the prompts in PIE-BENCH. The average caption length increases from 8 to 68. We bold the best performances. **ConceptAligner** outperforms SANA and SANA-Finetune.

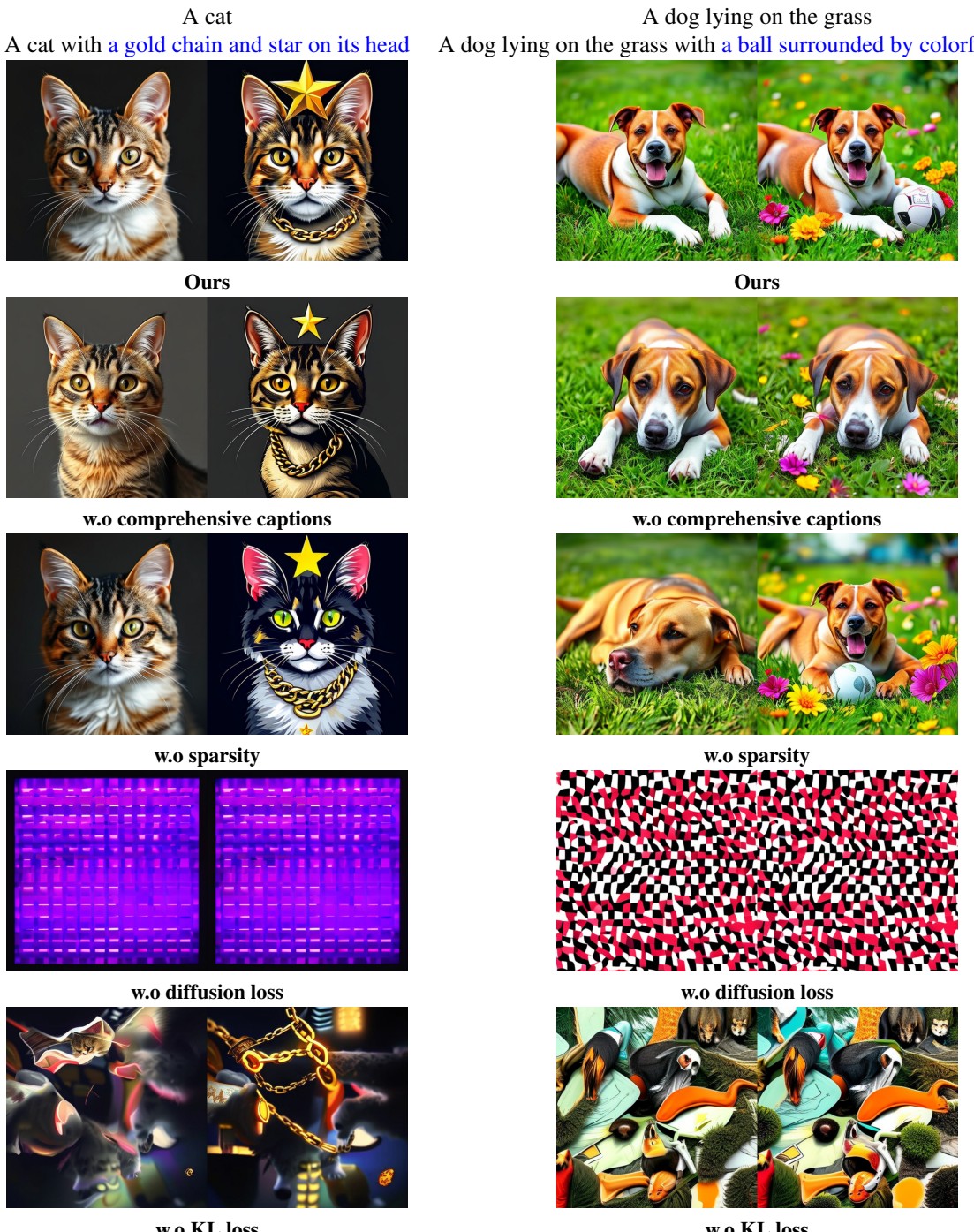

Figure 6: **Qualitative ablation results.** The five rows correspond to **ConceptAligner**, **ConceptAligner** without comprehensive captions, **ConceptAligner** without sparsity, **ConceptAligner** without diffusion loss, and **ConceptAligner** without KL loss. We can observe that **ConceptAligner** is robust to the caption quality degradation. Sparsity effectively minimizes unnecessary changes in each pair. The model cannot follow text prompts without diffusion losses and KL losses.

| Method | Sampling Speed (seconds/image) |
|---|---|
| SD3.5-M | 1.56 |
| SD3.5-L | 2.29 |
| FLUX-S | 0.63 |
| FLUX-D | 4.72 |
| SANA | 0.58 |
| **ConceptAligner** | 0.48 |

Table 6: **Sampling efficiency comparison on a H100 GPU.** Based on SANA, we employ three lightweight networks during training and only textual and concept networks are used for text-to-image generation. Each model contains 6 lightweight attention blocks. To further reduce the training and inference cost, we use 64 textual tokens for sampling. Compared to SANA, which employs 300 tokens (0.58 seconds/per sample), our method is the fastest with 0.48 seconds to sample an image.

| Setting | Method | CLIP-I ↑ | LPIPS ↓ | CLIP-T↑ | DINO ↑ |
|---|---|---|---|---|---|
| | BlendedDiffusion (Avrahami et al., 2022) | 0.837 | 0.377 | **0.303** | 0.562 |
| | Pix2pix-zero (Parmar et al., 2023) | 0.777 | 0.434 | 0.282 | 0.472 |
| Unpaired Images | Plug-and-Play (Tumanyan et al., 2023) | 0.887 | 0.324 | 0.301 | 0.705 |
| | PnpInv (Ju et al., 2024) | 0.892 | **0.313** | 0.301 | 0.720 |
| | LEDITS-SDXL (Tsaban & Passos, 2023) | 0.878 | 0.343 | 0.299 | 0.701 |
| | RF-Inversion (Wang et al., 2024a) | 0.906 | 0.427 | 0.285 | 0.737 |
| | Fireflow-FLUX (Deng et al., 2024) | 0.891 | 0.316 | 0.295 | 0.725 |
| Paired Images | Instruct-Pix2pix (Brooks et al., 2023) | 0.878 | 0.356 | 0.287 | 0.717 |
| Unpaired Images | **ConceptAligner** | **0.917** | 0.314 | 0.288 | **0.782** |

Table 7: **Evaluation results on real-world image editing dataset**. We use the source images, source and target prompts in PIE-BENCH dataset. Given a source image and source prompt, we apply standard diffusion inversion to obtain the initial noise of the diffusion process. We input the noise and the target prompt to produce the edited image. Across all metrics, **ConceptAligner** is superior or comparable to baseline methods, including those trained on expensive paired editing data. We bold the best performances.

Some palm trees and other plants are sitting on a highway overpass on a cloudy day.

Some palm trees and other plants are sitting on a highway overpass on a cloudy day with a double rainbow in the sky.

| FLUX-D | SANA | **ConceptAligner** |
|---|---|---|
| 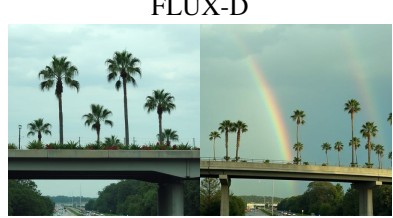 | 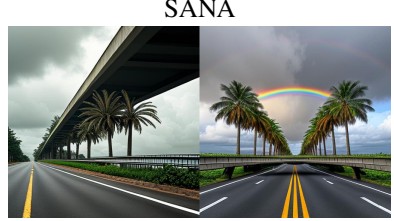 | 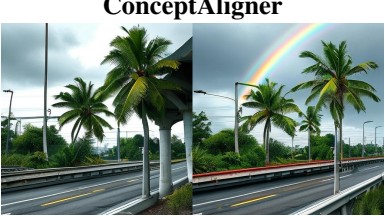 |

A red and yellow double decker bus on a small narrow street.

A Russian icon painting of a red and yellow double decker bus on a small narrow street.

| FLUX-D | SANA | **ConceptAligner** |
|---|---|---|
| 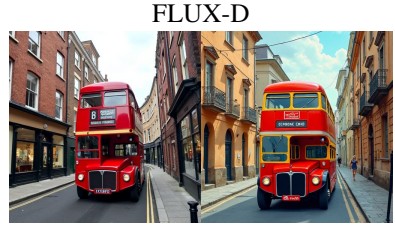 | 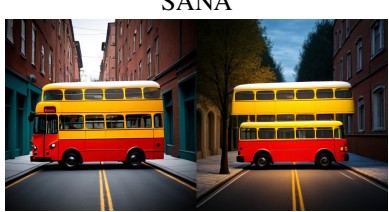 | 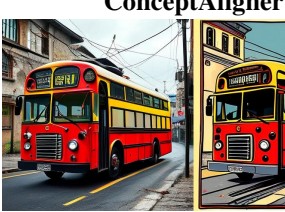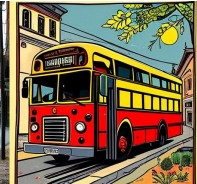 |

A woman in a black and white striped shirt has her arm reaching out.

A woman in a red and white striped shirt has her arm reaching out.

| FLUX-D | SANA | **ConceptAligner** |
|---|---|---|
| 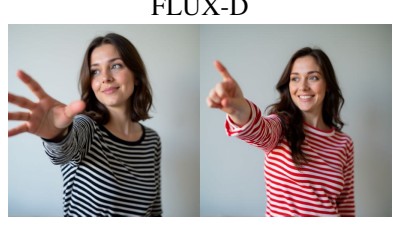 | 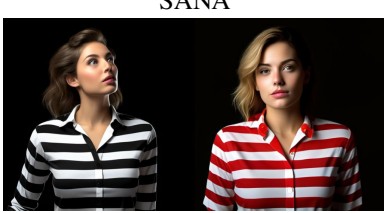 | 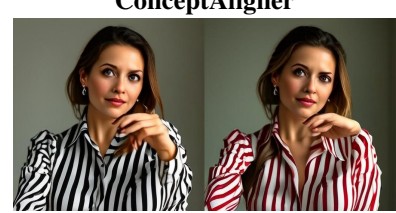 |

A giraffe walking in front of a lake.

A giraffe walking in front of a lake with a tree next to it.

| FLUX-D | SANA | **ConceptAligner** |
|---|---|---|
| 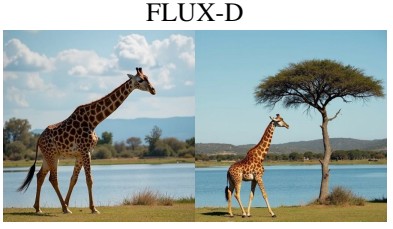 | 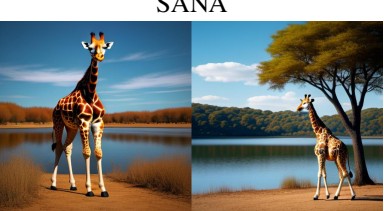 | 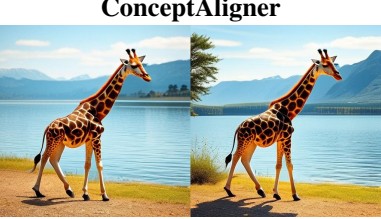 |

A couple of birds sitting on branches outside.

A couple of birds sitting on branches in a cage.

| FLUX-D | SANA | **ConceptAligner** |
|---|---|---|
| 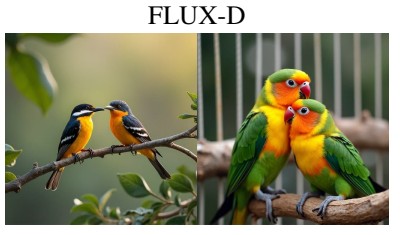 | 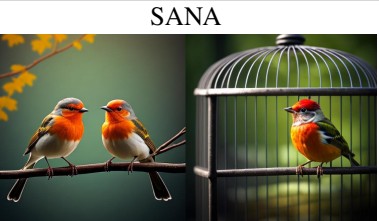 | 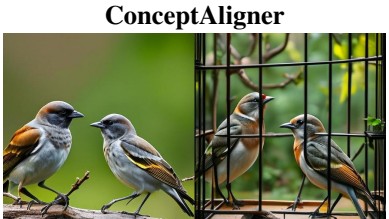 |

Figure 7: **Examples of controllable text-to-image generation comparisons.**

A cat.
A cat, wearing crown, smiling.

| SANA | **ConceptAligner** |
|---|---|

A dog, sketch.
A dog, watercolor, mouth opening.

| SANA | **ConceptAligner** |
|---|---|

A cartoon panda eating bamboo.
A cartoon panda eating bamboo, flowers around, butterflies around.

| SANA | **ConceptAligner** |
|---|---|

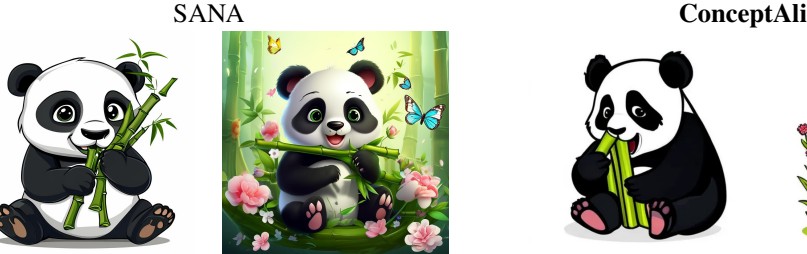

A beach.
A beach, a tree, a dog, a boat in the ocean, vangogh style.

| SANA | **ConceptAligner** |
|---|---|

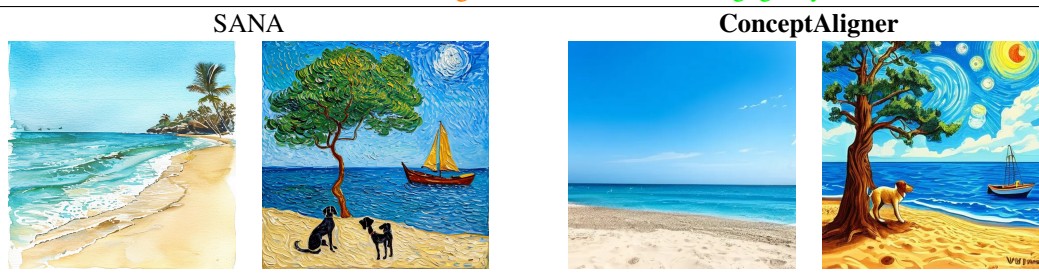

Figure 8: **Multiple concepts editing comparisons.** We use two prompts that differ by multiple concepts to generate paired images. Baseline SANA creates unnecessary changes, whereas **ConceptAligner** can preserve all desirable information, e.g., the dog's shape in the second row, the panda's pose in the third row, and the beach's layout in the last row. **ConceptAligner** can support simultaneously editing four concepts.

| SANA | SANA-Finetune | **ConceptAligner** |
|------|---------------|--------------------|

A dog is jumping.
A dog is sleeping.

A cat is swimming.
A cat is jumping.

An octopus is in a frozen tunddra.
An octopus is in a moonlit beach.

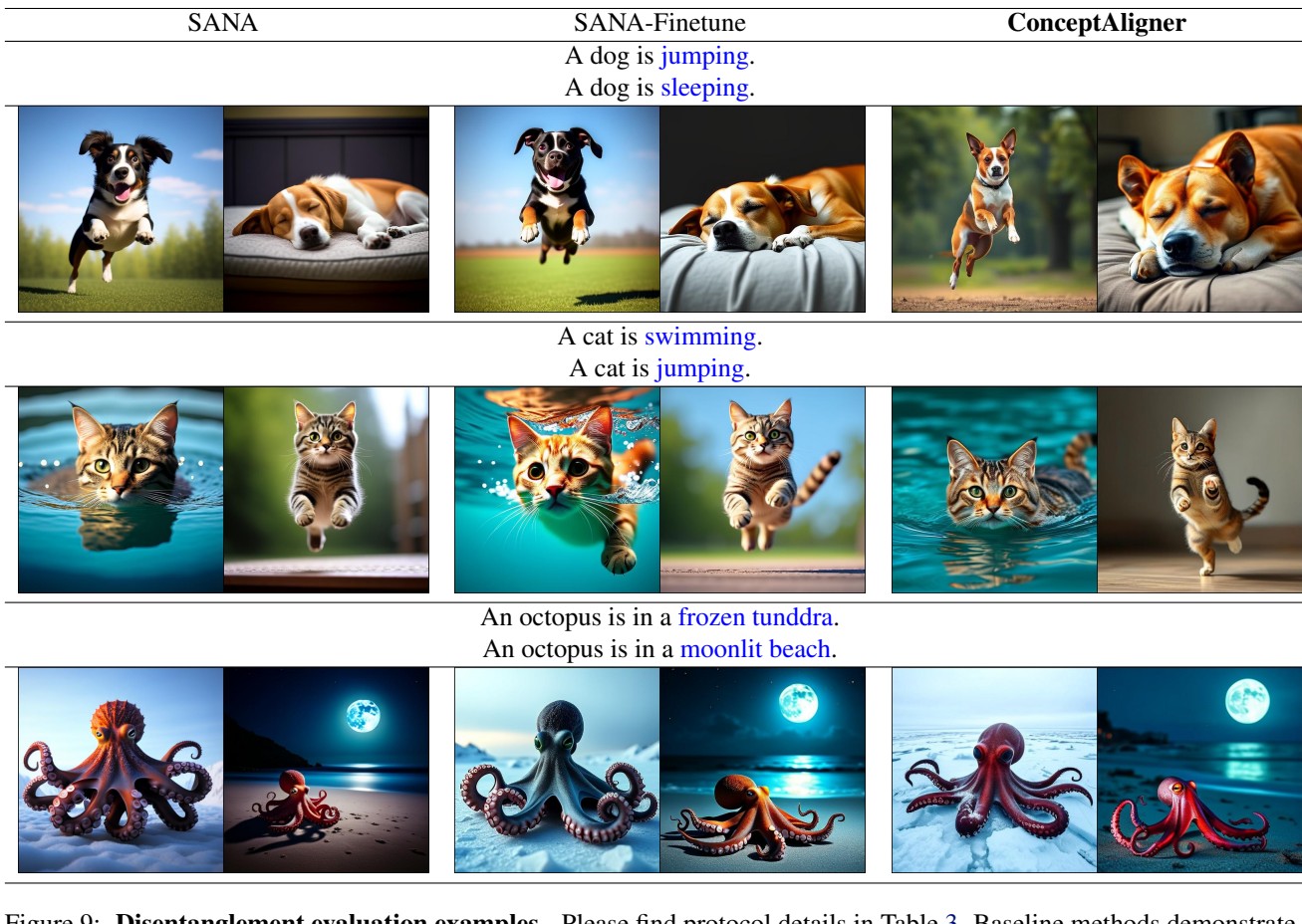

Figure 9: **Disentanglement evaluation examples.** Please find protocol details in Table 3. Baseline methods demonstrate inadequate preservation of subject identity during action or background modifications, our approach successfully maintains subject consistency throughout these transformations.

| Input | Plug-and-Play | PnpInv | InstructPix2pix | **ConceptAligner** |
|-------|---------------|--------|-----------------|---------------------|

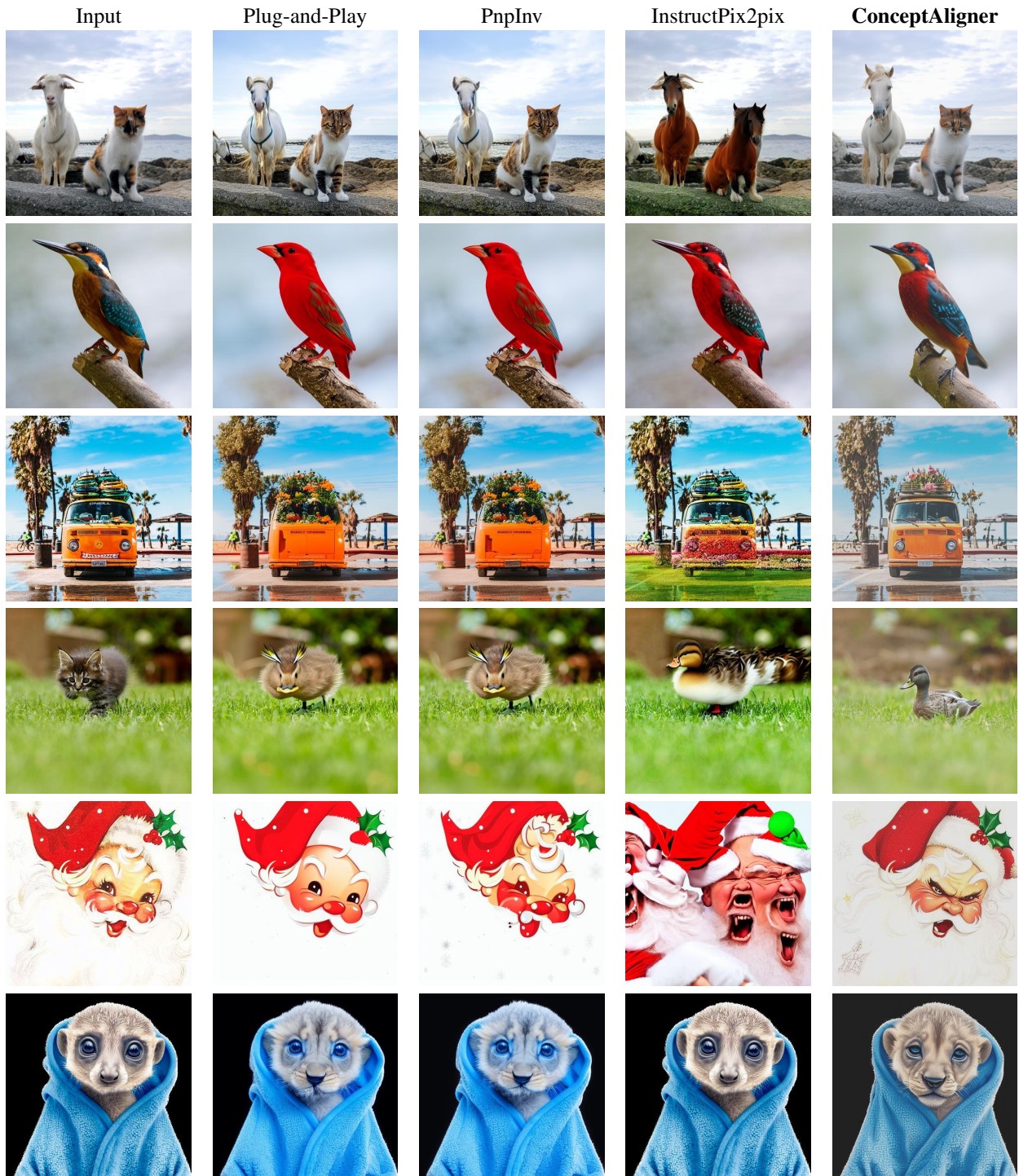

Figure 10: **Real-world image editing comparisons.** Target prompts: *photo of a horse and a cat standing on rocks near the ocean*, *a red bird standing on a branch*, *an orange van with flowers on top*,*the Christmas illustration of a Santa's angry face*, *a lion puppy wrapped in a blue towel*. The experimental details are provided in Table 7. **ConceptAligner** correctly follows the target prompt while preserving the subject identity.

