# OpenReview forum: "Learning Vision and Language Concepts for Controllable Image Generation"
_ICML.cc/2025/Conference — ICML 2025 poster_

### Official Review · Reviewer_pEXE · 2025-03-11

**Overall Recommendation:** 2

**Summary:**

This paper explores the theoretical foundations of concept learning for aligning atomic vision and language concepts, with applications in controllable text-to-image (T2I) generation. The authors formulate concept learning as a latent variable identification problem and propose a novel theoretical framework that guarantees component-wise identifiability under nonparametric conditions. The proposed model, ConceptAligner, explicitly disentangles atomic textual and visual concepts and ensures sparse connections between them. The authors demonstrate the effectiveness of ConceptAligner in controllable image generation tasks, showing improved interpretability and controllability compared to state-of-the-art methods.

**Claims And Evidence:**

Partially, the experiments may be less convincing to support the claims, please see the following parts.

**Essential References Not Discussed:**

N/A

**Experimental Designs Or Analyses:**

1. More human evaluations could further validate the improvements in interpretability and user control.

2. The paper states that learned text and visual concept interactions are implemented based on the causal graph $G^{t2i}$. However, in P3, line 157, the authors mention that the proposed framework can "capture statistical dependence." This contradicts the goal of causal graph discovery, as statistical dependence may introduce spurious correlations, which are undesirable in causality learning.

3. The model assumes that text descriptions provide sufficient variability for concept identification, which may not always hold in real-world datasets with ambiguous or incomplete captions. Moreover, the paper lacks details of implementation and experiments, including but not limited to the training dataset, comparison of text-based editing results,  the original prompts used for image generation, etc.

4. The comparative experiments may be unfair and less convincing. Specifically, the authors only compare the proposed model against standard text-to-image generation models but not against existing controllable text-to-image generation and image editing models. Moreover, is the controllable defined in this paper consistent with prior work on controllable T2I models?

5. The generated or edited images are not presented when modifying multiple concepts.

6.The paper lacks a detailed illustration of the learned concepts. While it shows text-to-image generation results after modifying a single concept, the atomicity among any pairwise concept is neither well-demonstrated nor validated. Additionally, how scalable is the learned concept? Can the model support online updates when encountering new concepts?

**Methods And Evaluation Criteria:**

Partially, please see the following parts.

**Other Comments Or Suggestions:**

Please see the above.

**Other Strengths And Weaknesses:**

More human evaluations could further validate the improvements in interpretability and user control.

The causal relationship in Equation (1) appears inconsistent with the description in Figure 1. Specifically, it seems that the correct formulation should be i = g^I(z^I) and vice versa.

The paper states that learned text and visual concept interactions are implemented based on the causal graph G^{t2i}. However, in P3, line 157, the authors mention that the proposed framework can "capture statistical dependence." This contradicts the goal of causal graph discovery, as statistical dependence may introduce spurious correlations, which are undesirable in causality learning.

The model assumes that text descriptions provide sufficient variability for concept identification, which may not always hold in real-world datasets with ambiguous or incomplete captions. Moreover, the paper lacks details of implementation and experiments, including but not limited to the training dataset, comparison of text-based editing results,  the original prompts used for image generation, etc.

The comparative experiments may be unfair and less convincing. Specifically, the authors only compare the proposed model against standard text-to-image generation models but not against existing controllable text-to-image generation and image editing models. Moreover, is the controllable defined in this paper consistent with prior work on controllable T2I models?

The generated or edited images are not presented when modifying multiple concepts.

ConceptAligner introduces additional computational complexity compared to conventional T2I models. The authors should report the training and inference costs of the proposed framework.

In Section 5.1, the authors state: "For text-based editing, we simply reuse the exogenous information ϵ of the previously generated image i." However, what if the input images come directly from real-world sources rather than being generated by the proposed model?

The paper lacks a detailed illustration of the learned concepts. While it shows text-to-image generation results after modifying a single concept, the atomicity among any pairwise concept is neither well-demonstrated nor validated. Additionally, how scalable is the learned concepts? Can the model support online updates when encountering new concepts?

**Questions For Authors:**

Please see the above

**Relation To Broader Scientific Literature:**

This paper aims to disentangle atomic concepts in text-to-image generation for enhanced interpretability and controllability.

**Theoretical Claims:**

I check the proof and it seems to be correct.

---

> ### Author Rebuttal · Authors · 2025-04-01
>
> Thank you for the valuable feedback. Please see our responses below and our uploaded results at https://anonymous.4open.science/r/ICML2025-F636/rebuttal.pdf.
>
> **1. Human evaluations.**
>
> Thank you for the nice suggestion. We have added the human evaluation results to the uploaded Figure 7. The human scores favor our method across all benchmarks.
>
> **2. Typos in Equation (1).**
>
> We have corrected the typo – thank you!
>
> **3. Capturing statistical dependence and spurious correlation.**
>
> Thank you for this thoughtful question. The statistical dependence among textual concepts $z ^{T}$ actually strengthens our framework rather than contradicts our causal goals. To clarify:
>
> Regardless of these dependencies, Theorem 4.4 guarantees we can disentangle individual textual concepts. If $z ^{T} _1$ represents "beach" and $z ^{T} _{2}$ represents "palm tree," we can identify them and intervene on "beach" without automatically changing "palm tree".
>
> At the same time, concepts may naturally exhibit statistical dependencies (like "beach" and "palm tree"). Our framework accommodates these dependencies, making it more general than models that impose independence constraints.
>
> **4. Sufficient variability in real-world datasets.**
>
> Great question! We have included experiments on training our model on short, coarse captions in uploaded Table 4. The negligible performance variation indicates our method’s robustness to the caption quality.
>
> Further, we have included the following discussion in our revision.
> ``In the paper, we give precise sufficient conditions – given adequately diverse data, we can achieve desirable component-wise identifiability. In general, we would expect this condition to be satisfied – standard text-image datasets (e.g., LAION) contain millions of captions, far exceeding the number of possible visual concepts. Additionally, we may follow existing methods (e.g., [1]) to employ vision-language models to generate higher-quality captions.’’
>
> **5. Implementation details.**
>
> Thank you for the thoughtful feedback. We have included implementation details in our revision.
>
> Training data: “We follow baseline SANA’s protocol to first generate 2 million images with Flux. 1 Schnell, and then apply QWEN2.0-VL for re-captioning. In total, we use 2M text-to-image data for finetuning.”
>
> Evaluation: “We use PIE-BENCH (Ju et al. 2024) for evaluation, which contains 700 editing instructions and corresponding input and output captions. We utilize the input and output captions to generate paired images and measure the similarity of each pair.”
>
> Original prompts: please see examples in uploaded Figure 2.
>
> **6. Comparative experiments & Is “controllable” consistent with prior work?**
>
> Prior work on “controllable T2I” often employs side information (e.g., edges) to control generation [2], whereas we use only text prompts to control atomic concepts. Thus, we didn’t compare with them. Thanks to your feedback, we have made this distinction explicit in our revision. We have included comparative experiments with recent image-editing methods in the uploaded Table 3. Our approach outperforms baseline approaches, including those trained on expensive paired editing data.
>
> **7. Images with multiple concepts modified & scalability of the learned concepts.**
>
> Thank you for the insightful question. We’ve included results on simultaneously modifying multiple concepts in the uploaded Figure 8. Our method manages to simultaneously edit up to four concepts, while the baseline quickly loses the subject identity.
>
> **8. Training and inference cost analysis.**
>
> We finetune our model (SANA backbone with LoRA) for 10000 steps, which takes around 12 hours on 8H100 GPUs. We provide the inference speed comparison in the uploaded Table 7. Thanks to our compact representation (64 vs. 300 tokens in SANA), our method is the fastest inference method (0.48s/image).
>
> **9. “What if the input images come directly from real-world sources?”**
>
> Given real-world images, we apply diffusion inversion to obtain the initial noise value, as in standard inversion-based models. We then feed the noise and the target prompt into our model for editing. The uploaded Table 3 and Figure 3 demonstrate that on real-world sources, our method is superior to or comparable with the baselines across all metrics.
>
> **10. Online updates for new concepts.**
>
> Great question – in that case, our model is flexible to incorporate a continuous learning strategy (e.g., dynamic memory expansion [3]) to absorb new concepts while retaining learned ones – thank you for suggesting this interesting direction.
>
> Please let us know whether your concerns are addressed. Thank you in advance!
>
> **References**
>
> [1] ShareGPT4V: Improving Large Multi-Modal Models with Better Captions. Chen et al. ECCV 2024. \
> [2] Adding Conditional Control to Text-to-Image Diffusion Models. Zhang et al. ICCV 2023. \
> [3] Online Task-Free Continual Generative and Discriminative Learning via Dynamic Cluster Memory. Ye et al. CVPR 2024.

---

### Official Review · Reviewer_Vot5 · 2025-03-13

**Overall Recommendation:** 3

**Summary:**

This paper explores concept learning by extracting interpretable "atomic concepts" from multimodal data (images and text) to support tasks like text-to-image (T2I) generation. It frames concept learning as a latent variable identification problem within a graphical model, establishing conditions for component-wise identifiability of atomic concepts under a flexible, nonparametric framework that handles both continuous and discrete modalities. Unlike prior work limited to block-wise identifiability or parametric constraints, the authors introduce ConceptAligner, a T2I model that learns disentangled textual and visual concepts with sparse connections.

## update after rebuttal
Thank you for your response. After reading the rebuttal, I keep my original score.

**Claims And Evidence:**

The claims in the paper are supported by quantitative results from ablation studies, such as Table 2 showing performance degradation without sparsity regularization. However, the paper provides only limited qualitative ablation results and lacks broader ablation experiment support. Furthermore, it does not conduct detailed analyses of the impact of individual loss functions (e.g., diffusion loss, KL divergence loss, and sparsity regularization loss), which restricts the comprehensiveness and persuasiveness of the evidence.

**Essential References Not Discussed:**

The paper omits key related works:

*Image Editing*: Xu et al.’s [12] InfEdit proposes inversion-free natural language image editing, highly relevant to this paper’s controllable generation goals, but is not cited.

*Causal Representation Learning*: Rajendran et al.’s [13] FCRL proposes a shift from causal to concept-based representation learning, directly related to this paper’s theoretical framework, but is not mentioned. Including these would offer a more comprehensive context for the contributions.

[12] Xu, S., Huang, Y., Pan, J., Ma, Z., and Chai, J. "Inversion-free Image Editing with Natural Language." Proceedings of the IEEE/CVF Conference on Computer Vision and Pattern Recognition (CVPR), 2024.

[13] Rajendran, G., Buchholz, S., Aragam, B., et al. "From Causal to Concept-Based Representation Learning." Advances in Neural Information Processing Systems, 37:101250-101296, 2024.

**Experimental Designs Or Analyses:**

I assessed the experimental design in Section 6:

*Design*: The experiments build on SANA [3], comparing against strong baselines like SD3.5-M/L and Flux.1-D/S, with ablation studies validating sparsity’s role.

*Issues*:

1) Lack of Downstream Task Comparisons: If the goal is to support downstream applications (e.g., controllable T2I generation or image editing), benchmarks against existing methods are essential. However, the paper lacks such comparisons, e.g., with recent image editing approaches. Adding relevant benchmark tests would significantly enhance the method’s practical validation.

2) Insufficient Dataset Description: The paper does not detail how the test dataset was processed, how many samples were selected, or how they were applied, limiting the reproducibility and generalizability assessment.

[3] Xie, E., Chen, J., Chen, J., Cai, H., Tang, H., Lin, Y., Zhang, Z., Li, M., Zhu, L., Lu, Y., et al. Sana: Efficient highresolution image synthesis with linear diffusion transformers. arXiv preprint arXiv:2410.10629, 2024.

**Methods And Evaluation Criteria:**

The proposed methods and evaluation criteria are appropriate for the controllable text-to-image (T2I) generation problem:

*Methods*: ConceptAligner’s architecture integrates a text encoder, image network, concept network, and diffusion transformer, aligning with the theoretical framework. The use of sparsity regularization and diffusion loss to enhance identifiability and generation quality is a reasonable design.

*Evaluation Criteria*: CLIP-I, LPIPS, and CLIP-T are standard metrics in T2I research, and testing on the paired prompt dataset from Ju et al. [1] aligns with the task goals. However, the evaluation metrics are relatively limited. For a study focused on image editing, emphasizing visual changes introduced by the method is crucial. For example, using DINO [2] to compute similarity between original and edited images could assess foreground and background consistency, providing a more comprehensive measure of transformation precision and quality. Adding such metrics would significantly enhance the evaluation’s rigor.

[1] Ju, X., Zeng, A., Bian, Y., Liu, S., and Xu, Q. Pnp inversion: Boosting diffusion-based editing with 3 lines of code. In The Twelfth International Conference on Learning Representations, 2024.

[2] Oquab, M., Darcet, T., Moutakanni, T., et al. "DINOv2: Learning Robust Visual Features without Supervision." Transactions on Machine Learning Research (TMLR), 2024.

**Other Comments Or Suggestions:**

1. The number of experimental results is insufficient, lacking additional comparative experiments.

2. Figures are difficult to interpret; for instance, the qualitative ablation in Figure 5 lacks labels distinguishing the original image from the one with sparsity regularization. I suggest adding annotations to improve readability.

3. All figures should be optimized for better comprehension of experimental outcomes.

**Other Strengths And Weaknesses:**

The framework diagram (Figure 2) is poorly designed, failing to clearly illustrate the training or inference pipeline. The operation of the concept network (R^C)—how it transforms external information and textual concepts into visual concepts—lacks intuitive explanation, with inputs and outputs not clearly labeled. I suggest optimizing the diagram to improve readability.

**Questions For Authors:**

*Article Structure and Experimental Completeness*: The overall structure feels rushed and incomplete, missing downstream task comparisons and parameter analyses. How would adding these affect the method’s evaluation?

*Dataset Selection Rationale*: Why were the current datasets chosen for testing, and what criteria justify this? The paper lacks explanation.

*Figures*: The figures (e.g., Figures 2 and 5) are hard to understand. How do you plan to improve them for clarity?

**Relation To Broader Scientific Literature:**

The paper relates to prior work in the following areas:

*Concept Learning*: Extends Kong et al.’s [4] single-modality work to multimodal scenarios.

*Causal Representation Learning*: Advances beyond block-wise identifiability by Yao et al. [5] and von Kügelgen et al. [6], aligning with Morioka & Hyvarinen’s [7][8] component-wise approach while relaxing parametric assumptions.

*T2I Generation*: Improves diffusion models by Rombach et al. [9] and ControlGAN by Li et al. [10], enhancing controllability. Its core contribution—component-wise identifiability with sparse multimodal connections—integrates theoretical identifiability from Khemakhem et al. [11] with T2I applications.

[4] Kong, L., Chen, G., Huang, B., et al. "Learning Discrete Concepts in Latent Hierarchical Models." The Thirty-eighth Annual Conference on Neural Information Processing Systems, 2024.

[5] Yao, D., Xu, D., Lachapelle, S., et al. "Multi-view Causal Representation Learning with Partial Observability." The Twelfth International Conference on Learning Representations, 2024.

[6] von Kügelgen, J., Sharma, Y., Gresele, L., et al. "Self-supervised Learning with Data Augmentations Provably Isolates Content from Style." arXiv preprint arXiv:2106.04619, 2021.

[7] Morioka, H. and Hyvarinen, A. "Connectivity-contrastive Learning: Combining Causal Discovery and Representation Learning for Multimodal Data." International Conference on Artificial Intelligence and Statistics, pp. 3399-3426. PMLR, 2023.

[8] Morioka, H. and Hyvarinen, A. "Causal Representation Learning Made Identifiable by Grouping of Observational Variables." Forty-first International Conference on Machine Learning, 2024.

[9] Rombach, R., Blattmann, A., Lorenz, D., et al. "High-Resolution Image Synthesis with Latent Diffusion Models." Proceedings of the IEEE/CVF Conference on Computer Vision and Pattern Recognition, pp. 10684-10695, 2022.

[10] Li, B., Qi, X., Lukasiewicz, T., and Torr, P. "Controllable Text-to-Image Generation." Advances in Neural Information Processing Systems, 32, 2019.

[11] Khemakhem, I., Kingma, D., Monti, R., and Hyvarinen, A. "Variational Autoencoders and Nonlinear ICA: A Unifying Framework." International Conference on Artificial Intelligence and Statistics, pp. 2207-2217. PMLR, 2020a.

**Theoretical Claims:**

I reviewed the proof of Theorem 4.4 (Appendix A). The proof reasons correctly under the assumption that Conditions 4.2 and 4.3 hold, and its logic is sound. However, the paper does not sufficiently justify the validity of each assumption or prove that these conditions hold generally. For instance, Condition 4.2-i (invertibility and smoothness of generating functions) is assumed true without explaining how it is verified in practice; Condition 4.3-4 (non-subset observed children) relies on sparsity but does not demonstrate its necessity. The lack of derivation or empirical validation of these assumptions undermines the credibility of the theoretical claims. I suggest the authors provide additional justification to improve understanding.

---

> ### Author Rebuttal · Authors · 2025-04-01
>
> We are grateful for your thorough assessment. Please find the responses below and our uploaded results at https://anonymous.4open.science/r/ICML2025-F636/rebuttal.pdf.
>
> **1. Detailed ablation analyses.**
>
> Thank you for your constructive feedback. We have added evaluations of the loss terms in the uploaded Table 4 and Figure 1. The sparsity regularization improves our model across almost all metrics. Without KL regularization, the exogenous variable $\epsilon$ contains excessive information about the input image, and the model ignores the text information as shown in Figure 1. The diffusion loss is the primary objective for the diffusion model, without which the model experiences negligible updates (Figure 1).
>
> **2. DINO similarities between original and edited images.**
>
> Great suggestion! We have included the DINO similarity in Table1, 2,3,4,6, and our revision. Our method consistently obtains the highest DINO similarity score across benchmarks.
>
> **3. Additional justifications for conditions.**
>
> Thank you for the thoughtful feedback. We have included the following discussion in our revision.
>
> Condition 4.2-i: ``The invertibility ensures the observed variables preserve all latent variables’ information. Otherwise, it would be impossible to recover these latent variables from observed variables.
> Practically, the high dimensionality of images often offers sufficient capacity to hold all information of $z ^{I}$, and human language is often a verbose articulation for concise, abstract textual concepts $ z ^{T} $, which makes this condition feasible.
> The smoothness allows us to use partial derivatives to establish identifiability, following prior work (Khemakhem et al., 2020a;b).
>
> Condition 4.3-4: “Condition 4.3-4 calls for sparse connections from the textual to the visual concepts. Consider concepts like "fur" and "ears" when describing a cat. These concepts should affect partially distinct visual features. If every visual feature triggered by "ears" was also triggered by "fur," these concepts aren't genuinely atomic and should be restructured.
> Theoretically, Condition 4.3-4 has been adopted in recent work (Kivva et al. 2021) and offers greater flexibility compared to alternatives in prior literature. For instance, prior work [1,2] assumes each $z ^{T} _{i}$ has at least one unique child $z ^{I} _{j}$, which is strictly stronger.
>
> We acknowledge that we only provide sufficient conditions that show the possibility of learning concepts. We completely agree that some conditions can be weakened. Nevertheless, developing necessary conditions for general problems is much more challenging, and we hope our work provides a foundation that can be iteratively refined by the research community.
>
> **4. Image-editing baselines.**
>
> Thank you for the helpful feedback. We have included image-editing baselines and datasets in the uploaded Table 3 and Figure 3. Our approach outperforms baseline approaches, including those trained on expensive paired editing data.
>
> **5. Computation efficiency analysis.**
>
> We finetune our model (SANA backbone with LoRA) for 10000 steps, which takes around 12 hours on 8H100 GPUs. We provide the inference speed comparison in the uploaded Table 7. Thanks to our compact textual representation (64 vs. 300 tokens in SANA), our method is the fastest inference method (0.48s/image).
>
> **6. Key related works.**
>
> Thank you for these valuable references. We have added the following discussion.
>
> “Xu et al. focus on designing attention maps where they replace the target attention map with the source map for a particular word. In contrast, we concentrate on developing superior conditioning representation. These two approaches are complementary and we leave investigating this synergy as future work.’’
>
> ``Rajendran et al. formulate concepts as affine subspaces of latent variables and provide identifiability guarantees for these subspaces. In contrast, we directly identify each latent variable, which enables us to directly control atomic aspects.’’
>
> **7. Figure optimization.**
>
> Thank you for the helpful feedback. We have re-designed the framework diagram (Figure 2) and annotated Figure 4 and Figure 5 – see uploaded Figure 4, 5, 6.
>
> **8. Additional downstream task comparisons and parameter analyses.**
>
> We have included additional ablation results as indicated in responses 1, 2, and 4. These results further highlight the advantages of our framework and validate our theoretical insights – thank you for the constructive suggestions!
>
> **9. Dataset Selection rationale.**
>
> We select PIE-BENCH (Ju et al. 2024) because it covers ten editing types and evenly distributed styles over four categories. This broad coverage can provide a thorough evaluation.
>
> Please let us know if you have any questions. We would be happy to discuss further!
>
> **References**
>
> [1] A Practical Algorithm for Topic Modeling with Provable Guarantees. Arora et al. ICML 2013. \
> [2] Identifiable Variational Autoencoders via Sparse Decoding. Moran et al. TMLR.

---

### Official Review · Reviewer_T6TG · 2025-03-13

**Overall Recommendation:** 3

**Summary:**

This paper introduces an Identification Theory for identifying atomic multimodal concepts. Leveraging this theory, the authors apply the method to controllable text-to-image generation. Both qualitative and quantitative evaluations have been conducted to assess the effectiveness of the proposed approach.

**Claims And Evidence:**

The details of how the proposed theory is applied to controllable text-to-image generation are inadequately explained, making it difficult to fully assess the validity of the claims. More details are listed in **Experimental Designs Or Analyses**.

**Essential References Not Discussed:**

N/A

**Experimental Designs Or Analyses:**

The experiments related to controllable text-to-image generation are not convincing for several reasons:

1. Essential details such as the dataset used for training, dataset size, and domain are not provided. Additionally, it is unclear whether the SANA is tuned alongside other components.
2. The token number for $z^T$ is set to 64. Is this sufficient to faithfully represent text information, especially for long text?
3. Details of the evaluation dataset are not provided.

**Methods And Evaluation Criteria:**

The proposed method aims to learn atomic concepts, but the metrics used do not evaluate the disentanglement among the learned concepts. It is recommended to incorporate additional metrics that specifically assess disentanglement to provide a more thorough evaluation.

**Other Comments Or Suggestions:**

There are several typos and minor issues:
1. In line 253, Our -> our
2. An extra space is present in the caption of Figure 2.
3. The first sentence may be repetitive with the second sentence in line 266.

**Other Strengths And Weaknesses:**

Strengths

1. Identifying atomic concepts for text-to-image models is both interesting and essential, as it enhances user control over the generation process.
2. The presented results show improved controllability in the text-to-image generation process.

Weaknesses

1. As discussed in the **Experimental Designs and Analyses** section, the experiments lack sufficient detail, undermining the effectiveness of the proposed theory.
2. Providing more results and metrics would be beneficial.

**Questions For Authors:**

My main concerns have been listed in the weaknesses part.

**Relation To Broader Scientific Literature:**

N/A

**Theoretical Claims:**

N/A

---

> ### Author Rebuttal · Authors · 2025-04-01
>
> We appreciate your valuable time and efforts dedicated to reviewing our work. Please find our responses below and the uploaded results at https://anonymous.4open.science/r/ICML2025-F636/rebuttal.pdf.
>
> **1. “It is recommended to incorporate additional metrics that specifically assess disentanglement to provide a more thorough evaluation.”**
>
> Thank you for this valuable suggestion. Unfortunately, most existing disentanglement metrics require access to ground truth latent variables (e.g.,  Mutual Information Gap [1]), which makes them not applicable to the text-to-image task where ground-truth latent variables are unavailable. Based on the definition of disentanglement, we design the following evaluation protocol to assess whether interventions on one latent factor would lead to unintended changes in other factors: We prompt ChatGPT to randomly name 10 animals, 10 actions, and 10 backgrounds. For each animal, we fix the animal's identity and edit its action and background, resulting in 200 original & edited pairs. We repeat over 10 random seeds and end up with 2000 pairs in total. For evaluation, we employ QWEN2.5-VL-Instruct7B to examine whether the editing retains the animal identity (subject consistency) and whether the targeted modification is achieved (prompt consistency). As you can see from uploaded Table 5 and Figure 9, our method achieves the highest subject consistency and prompt consistency scores simultaneously against SANA and finetuned SANA, demonstrating its superior disentanglement capability.
>
>
> **2. Details on the training dataset, dataset sizes, domains, and whether SANA is tuned with other components.**
>
> Thank you for pointing this out. As you suggest, we’ve included a dedicated section in the appendix to cover these details.
> “SANA employs around 30 million text-to-image paired data to train the model. Unfortunately, the data is not publicly available. Therefore, we follow SANA’s protocol to first generate 2 million images with Flux. 1 Schnell, and then apply QWEN2.0-VL for re-captioning. In total, we use 2M text-to-image data for finetuning.”
>
> In our model, the SANA backbone is fine-tuned via LoRAs. Thanks to your suggestion, we have included results on finetuning SANA via LoRAs on our data. As shown in Table 4, our model maintains the leading margin against SANA-Finetune, showcasing our approach’s effectiveness.
>
> **3. ``The token number for $z ^{T} $ is set to 64. Is this sufficient to faithfully represent text information, especially for long text?’’**
>
> Thank you for the insightful question. In light of your question, we have added comparative experiments on long captions. Specifically, we expand the short prompts in PIE-BENCH [2] to longer captions with QWEN2.5-Instruct-32B.  We can see from Table 6 that our method outperforms SANA and SANA-Finetune (which uses a token number of 300) while enjoying faster training and inference, thanks to our compact representation (a token number of 64).
>
> **4. ``Details on the evaluation dataset.’’**
>
> Thank you for raising this point. We have included the following details in the appendix.
>
> “We use the benchmark PIE-BENCH [2], which contains 700 editing instructions and corresponding input and output captions. We utilize the input and output captions to generate paired images and measure the similarity of each pair.”
>
> **5. ``More results.’’**
>
> Thanks to your feedback, we have included two new sets of evaluation results: 1) In Table 2 and Figure 2, we present additional experiments on the EMU-edit test set [3], which consists of 3,589 paired prompts and encompasses seven common editing types: background alteration (background), comprehensive image changes (global), style modification (style), object removal (remove), object addition (add), localized modifications (local), and color/texture changes (texture). We can observe that our approach either outperforms or is comparable with baseline methods across all metrics, demonstrating its effectiveness. 2) We also added real-image editing results in Table 3 and Figure 3. Our method shows superior performance against image editing baselines.
>
>
> **6. Typos.**
>
> We have corrected all these typos in our manuscript – thank you so much for the helpful feedback!
>
>
> We were wondering if we have addressed all your concerns. Please let us know if there is anything we could further discuss – your further feedback would be greatly appreciated.
>
>
> **References**
>
> [1] Isolating sources of disentanglement in variational autoencoders. Chen et al. NeurIPS 2018. \
> [2] PnP Inversion: Boosting Diffusion-based Editing with 3 Lines of Code. Ju et al. ICLR 2024. \
> [3] Emu Edit: Precise Image Editing via Recognition and Generation Tasks. Sheynin et al. CVPR 2024.

---

> > ### Comment · Reviewer_T6TG · 2025-04-07
> >
> > Thank you to the authors for their efforts. After reading the rebuttal and comments from other reviewers, most of my concerns regarding the experimental details have been addressed. However, as mentioned by other reviewers, the paper lacks a fair comparison with controllable image generation methods. Although the authors have added comparisons with some image editing methods, such as Pix2pix-zero, BlendedDiffusion, and Instruct-Pix2pix, these methods are based on older T2I backbones (e.g., SD 1.5 instead of SANA or FLUX), which may make the comparisons unfair. It would be beneficial to include comparisons with more recent editing methods (e.g., LEDITS++[1] on SDXL and RF Inversion[2] on FLUX or other methods) to demonstrate the effectiveness of the proposed method. Given the limited time available, adding these comparisons using the examples presented in Fig. 3 of the rebuttal is acceptable. I'd like to adjust my rating accordingly.
> >
> > [1] LEDITS++: Limitless Image Editing using Text-to-Image Models
> > [2] Semantic Image Inversion and Editing using Rectified Stochastic Differential Equations
> >
> > ----
> >
> > Update:
> >
> > Thank you to the authors for the prompt reply. My concerns have been addressed, and I will update my rating to 3.

---

> > > ### Author Response · Authors · 2025-04-08
> > >
> > > Thank you for your prompt and informative feedback! We completely agree that comparing more recent methods would strengthen our paper. We immediately dedicated time to implementing these comparisons. We have included the quantitative results in the updated Table 3 and visual comparisons in the newly added Figure 10 and 11 at https://anonymous.4open.science/r/ICML2025-F636/rebuttal.pdf.
> > >
> > > In light of your recommendation, we’ve conducted the comparison with LEDITS++ [1] on SDXL, RF inversion [2] on FLUX, and an even more recent method FireFlow [3] which employs a numerical solver for the ODEs underlying rectified flow models for inversion and editing. The quantitative results are as follows (we have also included these results in Table 3 in the [rebuttal PDF](https://anonymous.4open.science/r/ICML2025-F636/rebuttal.pdf) and our revision):
> > >
> > > | Method | CLIP-I ↑ | LPIPS ↓ | CLIP-T ↑ | DINO ↑ |
> > > |--------|---------|---------|---------|--------|
> > > | LEDITS-SDXL-CVPR2024 | 0.878 | 0.343 | **0.299** | 0.701 |
> > > | RF-Inversion-FLUX-ICLR2025 | 0.906 | 0.427 | 0.285 | 0.737 |
> > > | Fireflow-FLUX, Arxiv, Dec10, 2024 | 0.891 | 0.316 | 0.295 | 0.725 |
> > > | Concept Aligner | **0.917** | **0.314** | 0.288 | **0.782** |
> > >
> > > Our method outperforms all baselines on three out of four metrics (CLIP-I, LPIPS, and DINO) while remaining competitive on CLIP-T. This demonstrates that our approach isn't just theoretically sound but delivers practical advantages over even the most recent methods that use modern T2I backbones.
> > >
> > > Following your suggestion, we have updated our [rebuttal PDF](https://anonymous.4open.science/r/ICML2025-F636/rebuttal.pdf) to include visual examples (Figure 10) using the same images from the rebuttal Figure 3.
> > > Across all examples, our method achieves the targeted edits while retaining the other elements untouched. For example, in the second row, while other methods either fail to correctly change the bird's color (RF-inversion, FireFlow) or distort its appearance (LEDITS++), our method successfully produces a red bird while preserving its original form and details.
> > >
> > > In addition, we've included comparison examples in Figure 11 to further demonstrate these advantages across various editing scenarios. In the first example, our approach successfully transforms the rabbit into a cat while maintaining image coherence, a task where the baseline methods struggled. The second example demonstrates our method's ability to introduce a "monster" element while preserving the woman's facial features, whereas LEDITS++ and FireFlow couldn't effectively render the monster concept, and RF-Inversion unfortunately altered the woman's appearance.
> > >
> > > These examples further support our quantitative findings and demonstrate the practical advantages of our approach in maintaining both editing fidelity and preserving unrelated image elements, even against state-of-the-art approaches using advanced T2I backbones.
> > >
> > > Thank you again for enabling us to strengthen this aspect of our paper! Your suggestion has helped us substantially improve the quality of our work. We hope these additional comparisons address your concerns and merit a reconsideration of your rating.
> > >
> > > **References**
> > >
> > > [3] FireFlow: Fast Inversion of Rectified Flow for Image Semantic Editing. Deng et al. https://arxiv.org/abs/2412.07517.

---

### Official Review · Reviewer_Eyu1 · 2025-03-14

**Overall Recommendation:** 3

**Summary:**

- This paper addresses the problem of learning atomic multimodal concepts by proving under certain nonparametric assumptions, it is possible to component-wise identify each textual concept and each visual concept.

- Guided by this theory, they propose ConceptAligner, a T2I model that explicitly learns discrete textual concepts and continuous visual concepts with a sparse bipartite graph between them.

- Empirically, the paper shows that ConceptAligner outperforms existing T2I methods on controllability and visual quality metrics.

**Claims And Evidence:**

1. **Claim:** Atomic multimodal concepts can be learned with component-wise identifiability under nonparametric assumptions. **Evidence:** The paper offers a formal theoretical framework (Section 4) culminating in Theorem 4.4

2. **Claim:** ConceptAligner outperforms standard text-to-image baselines in controllable generation. **Evidence:** The paper compares ConceptAligner with Stable Diffusion, FLUX, and SANA, showing better performance quantitively and qualitatively  .

**Essential References Not Discussed:**

No essential references missed.

**Experimental Designs Or Analyses:**

1. The paper primarily uses a paired prompt scenario from u et al. (2024) and CLIP/LPIPS scores to measure how well the model modifies certain image attributes and keeps other attributes.

2. The paper also presents a ablation study showing sparse text-to-visual coupling is needed for robust control.

3. However, the paper only show results from one dataset, lacking extensive experiments on various domains.

**Methods And Evaluation Criteria:**

1. The paper proposes a text network, an image network, a concept network, and a conditional diffusion model rendering out the
visual representation to image.

2. The proposed ConceptAligner and other SOTA methods are evaluated with CLIP scores, LPIPS scores, and some qualitative examples.

3. Overall, the proposed methods and evaluation criteria are fairly standard in controllable generation and make sense.

**Other Comments Or Suggestions:**

See previous sections

**Other Strengths And Weaknesses:**

**Strengths**
1. The paper shows a nonparametric approach that can handle mixed discrete and continuous latent variables.
2. The generation results is impressive compared to other SOTA models.

**Weaknesses**
1. The paper only show results from one dataset, lacking extensive experiments on various domains.
2. Real-world text–image datas might sometimes violate the “sufficient variability” condition. The paper will be more sound if it shows how robust the method is if the textual descriptions are not comprehensive or are repetitive.

**Questions For Authors:**

See previous sections

**Relation To Broader Scientific Literature:**

- The paper situates itself at the intersection of multimodal representation learning, causal representation learning, and conditional generation.

**Theoretical Claims:**

1. The paper claims under certain invertibility, smoothness, conditional independence, non-degeneracy, and sparsity assumptions, both discrete textual concepts and continuous visual concepts can be identified component-wise from text–image pairs in Theorem 4.4.

2. I found no obvious flaws in the claim and derivations.

---

> ### Author Rebuttal · Authors · 2025-04-01
>
> Thank you for the time dedicated to reviewing our paper, the insightful comments, and valuable feedback. Please see our point-by-point responses below and the uploaded results at https://anonymous.4open.science/r/ICML2025-F636/rebuttal.pdf.
>
> **1. Experiments on various domains.**
>
> Thank you for your constructive comment. In light of your comment, we have included experiments on the EMU-edit testset [1] in our manuscript and uploaded Table 2.  The EMU-edit testset contains 3589 paired prompts and covers 7 common editing types, including background alteration (background), comprehensive image changes (global), style alteration (style), object removal (remove), object addition (add), localized modifications (local), and color/texture alterations (texture). We also provide generated samples in the uploaded Figure 2.  In addition to the controllable text-to-image generation task,  we also evaluate our method on real-world image editing tasks against image editing baselines in uploaded Table 3 and Figure 3. Across all metrics, our method is superior or comparable to the baselines, showcasing the effectiveness of our framework.
>
>
> **2. “Real-world text-image datasets might violate the `sufficient variability’ condition. Add experiments to show how robust the method is.”**
>
> Thanks to your suggestion, we have included in our manuscript and the uploaded Table 4 experiments on short, coarse captions to demonstrate the robustness of our approach.
> In our main experiments in the submission, we follow our baseline SANA’s [2] protocol to generate the training data: we first randomly sample 2 million short prompts from DiffusionDB [3] to generate 2 million images with Flux. 1 Schnell, and then apply QWEN2.0-VL to generate detailed captions. Here, to assess our model’s robustness to the text quality, we replace the detailed captions with original short prompts for model training. We can observe that across all metrics, the text-quality degradation makes negligible impacts on our method, demonstrating its robustness to text quality changes.
>
> To further address your concern, we have included the following discussion in our revision.  `` In the paper, we give precise sufficient conditions – given adequately diverse data,  we can achieve desirable component-wise identifiability. In general, we would expect this condition to be satisfied – standard text-image datasets (e.g., LAION) contain millions of captions, far exceeding the number of possible visual concepts. Additionally, we may follow existing methods (e.g., [3,4,5]) to employ vision-language models to generate higher-quality captions.
> Even if this condition is not met, oftentimes other natural properties can be leveraged to greatly weaken this condition. For instance, if the generating function $g ^{I}$ is simple or sparse, less variability would be needed to guarantee the identifiability (e.g., [6]). Such sparsity is often encouraged implicitly or explicitly in generative models (e.g., sparse attention patterns). A thorough theoretical investigation into combining these properties is an interesting problem, which we leave as future work.''
>
> Please let us know if there are any further concerns, and we are more than happy to address them in the following stage.
>
> **References**
>
> [1] Emu Edit: Precise Image Editing via Recognition and Generation Tasks. Sheynin et al. CVPR 2024. \
> [2] SANA: Efficient High-Resolution Image Synthesis with Linear Diffusion Transformers. Xie et al. ICLR 2025. \
> [3] DiffusionDB: A Large-scale Prompt Gallery Dataset for Text-to-image Generative Models." Wang et al. arXiv. \
> [4] ShareGPT4V: Improving Large Multi-Modal Models with Better Captions. Chen et al. ECCV 2024. \
> [5] Improving Image Generation with Better Captions. Betker et al. Technical report. \
> [6] Synergy Between Sufficient Changes and Sparse Mixing Procedure for Disentangled Representation Learning. Li et al. ICLR 2025.

---

### Decision · Program_Chairs · 2025-05-01

**Decision:**

Accept (poster)

**Comment:**

This paper addresses the problem of learning atomic multimodal concepts. The final recommendations are 1 "Weak reject" and 3 "Weak accept". The reviewers generally agree that
- the problem is interesting and important,
- the proof of the proposed theory for identifying atomic multimodal concepts sounds,
- the ConceptAligner outperforms exisiting T2I methods on controllability

Their major concerns include
- limited evaluation and limted evalution metrics,
- insufficient description of the dataset and experiments,
- poor illurations and explanations,
- lack of a fair comparison with controllable image generation methods,
- validity of each assumption has not been sufficiently justified,
- robustness unclear when assumptions are not met

In their rebuttal, the authors have made great effort to address the reviewers concerns by providing additional experimental results (in terms of more baselines, dataset, and evaluation metrics) and improved over their initial submission. I recommend "Weak accept".